# FcRn-Driven Nanoengineered Mucosal Vaccine with Multi-Epitope Fusion Induces Robust Dual Immunity and Long-Term Protection Against *Brucella*

**DOI:** 10.3390/vaccines13060567

**Published:** 2025-05-26

**Authors:** Tingting Tian, Yuejie Zhu, Kaiyu Shang, Huidong Shi, Ruixue Xu, Mingzhe Li, Fuling Pu, Junyu Kuang, Jianbing Ding, Fengbo Zhang

**Affiliations:** 1State Key Laboratory of Pathogenesis, Prevention and Treatment of High-Incidence Diseases in Central Asia, The First Affiliated Hospital of Xinjiang Medical University, Graduate School, Clinical Laboratory Diagnostics, Urumqi 830000, China; 18699024774@163.com (T.T.); xjmusky@163.com (K.S.); 18160215616@163.com (H.S.); dingjb1234@aliyun.com (J.D.); 2Reproductive Fertility Assistance Center, The First Affiliated Hospital of Xinjiang Medical University, Urumqi 830000, China; 3Department of Medical Laboratory Technology, School of Medicine, Xinjiang Medical University, Urumqi 830000, China; 19846908091@163.com (R.X.); 17394951165@163.com (M.L.); py207663829@163.com (F.P.); kjy20041364@163.com (J.K.); 4Department of Clinical Laboratory, The First Affiliated Hospital of Xinjiang Medical University, Urumqi 830000, China

**Keywords:** FcRn: neonatal Fc receptor, MEV: multi-epitope vaccine, mucosal vaccine

## Abstract

Background: Brucellosis poses a significant public health challenge, necessitating effective vaccine development. Current vaccines have limitations such as safety concerns and inadequate mucosal immunity. This study aims to develop an FcRn-targeted mucosal *Brucella* vaccine by fusing the human Fc domain with *Brucella*’s multi-epitope protein (MEV), proposing a novel approach for human brucellosis prevention. Methods: The study developed a recombinant antigen (h-tFc-MEV) through computational analyses to validate antigenicity, structural stability, solubility, and allergenic potential. Molecular simulations confirmed FcRn binding. The vaccine was delivered orally via chitosan nanoparticles in murine models. Immunization was compared to MEV-only immunization. Post-challenge assessments were conducted to evaluate protection against *Brucella* colonization. Mechanistic studies investigated dendritic cell activation and antigen presentation. Results: Computational analyses showed that the antigen had favorable properties without allergenic potential. Molecular simulations demonstrated robust FcRn binding. In murine models, oral delivery elicited enhanced systemic immunity with elevated serum IgG titers and amplified CD4+/CD8+ T-cell ratios compared to MEV-only immunization. Mucosal immunity was evidenced by significant IgA upregulation across multiple tracts. Long-term immune memory persisted for six months. Post-challenge assessments revealed markedly reduced *Brucella* colonization in visceral organs. Mechanistic studies identified FcRn-mediated dendritic cell activation through enhanced MHC-II expression and antigen presentation efficiency. Conclusions: The FcRn-targeted strategy establishes concurrent mucosal and systemic protective immunity against *Brucella* infection. This novel vaccine candidate shows potential for effective human brucellosis prevention, offering a promising approach to address the limitations of current vaccines.

## 1. Introduction

Brucellosis, a globally prevalent zoonosis caused by the facultative intracellular pathogen *Brucella* spp., imposes dual burdens on animal husbandry and public health. Economically, it results in substantial livestock production losses due to reproductive failures, such as abortion and infertility, as well as decreased milk yield [1]. From a public health perspective, the pathogen transmits to humans via mucosal adhesion mechanisms through contaminated dairy products, occupational exposure to infected animals, or percutaneous injuries [2,3]. Clinical manifestations range from undulant fever and night sweats to severe complications, including endocarditis (observed in 2% of chronic cases) and neurobrucellosis (in 5% of untreated infections) [4,5].

Current prevention strategies predominantly rely on live-attenuated vaccines, such as *B. abortus* S19 and *B. melitensis* Rev.1, which have proven instrumental in national eradication programs [6,7,8]. Nevertheless, two unresolved safety concerns limit their application [9]: (1) residual virulence that poses infection risks to immunocompromised hosts [10]; (2) occupational transmission hazards through accidental needle-stick injuries among veterinary workers [11,12]. These limitations underscore the critical need for novel vaccine platforms that balance protective efficacy with biosafety.

The mucosal route is predominant in *Brucella* infection, making intranasal or oral mucosal vaccines a strategic intervention to reinforce the primary defense barrier at pathogen entry sites [13]. By targeting mucosa-associated lymphoid tissues (MALT), these vaccines induce localized immune responses characterized by the production of secretory IgA (sIgA), which neutralizes pathogens at mucosal surfaces while simultaneously activating systemic cellular immunity through the cross-priming of CD8+ cytotoxic T lymphocytes (CTLs) and CD4+ T helper cells [14]. Specifically, CTLs execute direct clearance of mucosal-adherent bacteria via cytolytic mechanisms, whereas CD4+ T cells enhance Th1-polarized immunity through the secretion of interferon-γ(IFN-γ), synergistically improving intracellular pathogen elimination [14,15].

Advances in immunoinformatics have further revolutionized vaccine design by enabling the computational prediction of conserved immunodominant epitopes in conjunction with structural vaccinology approaches. Such epitope-based vaccines offer three transformative advantages over conventional platforms: (1) accelerated development cycles through in silico epitope screening and 3D structural simulations, (2) enhanced thermostability that circumvents cold-chain logistics, and (3) modular antigen designs that allow for rapid customization against evolving pathogens [16]. This integrated approach not only aligns immune interventions with natural infection routes but also addresses critical limitations of traditional vaccine paradigms.

The VirB10 protein, a core component of *Brucella*’s type IV secretion system (T4SS), exhibits unique structural and functional properties that underscore its potential as a vaccine candidate. Its modular architecture—comprising an N-terminal cytoplasmic domain, a transmembrane helix, a flexible linker region, and a globular C-terminal domain (CTD)—enables the integration of multiple protein partners and the coordination of transmembrane signaling [4,17]. Crucially, the *Brucella* T4SS demonstrates species-specific adaptations in host tropism and pathogenesis, positioning VirB10 as a precise target for species-discriminative vaccine design. Experimental studies have validated its immunogenicity, showing that VirB10 elicits robust humoral immunity, characterized by IgG titers, and Th1-polarized cellular responses, which include IFN-γ-secreting CD4+ T cells and cytotoxic CD8+ T lymphocytes, collectively conferring protection against intracellular bacterial persistence [4,17,18]. Concurrently, the highly conserved outer membrane protein Omp25 plays a dual role in *Brucella* pathogenesis and immune evasion. Structurally, Omp25 stabilizes bacterial envelope integrity through β-barrel folding and lipoprotein anchoring, which are prerequisites for resistance to environmental stress [19,20]. Functionally, it subverts host immunity by suppressing TNF-α production in infected macrophages via interference with the TLR4/NF-κB pathway, thereby promoting intracellular survival [21]. Structural analyses reveal that Omp25 engages host cell receptors (e.g., fibronectin-binding integrins) through surface-exposed loops, facilitating bacterial adhesion and niche establishment [22]. Consistent with its role in virulence, Omp25-deficient strains exhibit attenuated infectivity in murine models, characterized by reduced splenic colonization and prolonged survival rates [23]. These attributes, combined with its immunodominant epitopes recognized across *Brucella* species, designate Omp25 as a prime candidate for subunit vaccine development, offering cross-protective potential against diverse *Brucella* serovars [24].

To overcome the limited efficacy of conventional vaccines against *Brucella*’s intracellular persistence [11,25,26], we developed an FcRn-targeted mucosal vaccine strategy leveraging bioinformatics-predicted virulence determinants. By engineering antigens fused to IgG Fc domains, this design capitalizes on FcRn-mediated transcytosis to achieve spatiotemporal optimization: (1) mucosal antigen enrichment through FcRn recycling extends tissue residency and systemic circulation half-life, ensuring prolonged immunogenic stimulation; (2) Fcγ receptor-driven dendritic cell internalization enhances antigen cross-presentation via concurrent MHC-I/II processing, eliciting synchronized CD4+ T follicular helper cell-driven B-cell maturation and CD8+ cytotoxic T lymphocyte activation. This dual-axis mechanism—synergizing spatial targeting precision with sustained antigen availability—establishes a multilayered defense system that surmounts traditional vaccine limitations, including inadequate mucosal barrier penetration, transient antigen exposure, and inefficient intracellular pathogen clearance, thereby providing coordinated humoral-cellular immunity against facultative intracellular pathogens like *Brucella* [27,28]. This study aimed to investigate the mucosal immunogenicity of h-tFc-MEV vaccine mediated by FcRn-targeting strategy, particularly focusing on the generation of antigen-specific antibodies in mucosal secretions.

## 2. Materials and Methods

### 2.1. Design and Analysis of Bioinformatics

#### 2.1.1. Physicochemical Property Analysis

The amino acid sequences of VirB10 and Omp25 were retrieved from the UniProt database 2024_04 (URL: https://www.uniprot.org; accessed on 13 March 2024). Subsequently, ProtParam 2.3.1 (URL: http://web.expasy.org/protparam/; accessed on 18 March 2024) was employed to analyze their physicochemical characteristics, including molecular weight, theoretical isoelectric point (pI), amino acid/atomic composition, extinction coefficient, estimated half-life, instability index, aliphatic index, and grand average of hydropathicity (GRAVY) [29].

#### 2.1.2. Homology Assessment

Sequence homology analysis was performed using NCBI BLASTP 2.14.0+ (URL: https://blast.ncbi.nlm.nih.gov/Blast.cgi?PAGE=Proteins; accessed on 2 April 2024) against the human proteome (TaxID: 9606) with an E-value threshold of 0.001 [30]. This comparative analysis aimed to exclude potential human homologs for both native proteins (VirB10 and Omp25) and the designed vaccine construct.

#### 2.1.3. Antigenicity Evaluation

VaxiJen 2.0 server (URL: http://www.ddg-pharmfac.net/vaxijen/VaxiJen/VaxiJen.html; accessed on 2 April 2024) was utilized for antigenicity prediction of candidate proteins and vaccine components. A species-specific threshold of 0.4 was applied for bacterial antigen classification. All evaluated peptides met the minimum length requirement (>5 amino acids).

#### 2.1.4. Allergenicity Screening

Potential allergenicity was assessed using AllerTOP v3.0 (URL: https://www.ddg-pharmfac.net/AllerTOP/; accessed on 2 April 2024), a machine learning-based platform employing amino acid composition and physicochemical properties. Peptide sequences exceeding five residues were systematically analyzed through this bioinformatic filter.

#### 2.1.5. Transmembrane Topology Prediction

TMHMM 2.0 (URL: https://services.healthtech.dtu.dk/service.php?TMHMM-2.0; accessed on 2 April 2024) was implemented to identify transmembrane helices and determine membrane-spanning orientations. This analysis guided subsequent epitope selection by excluding hydrophobic transmembrane domains from potential antigenic regions.

#### 2.1.6. Signal Peptide Identification

SignalP 5.0 (URL: https://services.healthtech.dtu.dk/service.php?SignalP-5.0; accessed on 2 April 2024) with neural network algorithms was used to detect N-terminal signal peptides. Predicted signal peptide regions (probability > 0.7) were excluded from epitope screening to focus on mature protein domains.

#### 2.1.7. Toxicity Profiling

The ToxinPred 1.0 (URL: https://webs.iiitd.edu.in/raghava/toxinpred/; accessed on 5 April 2024) approach (alignment-based and machine learning methods) was applied to evaluate candidate epitopes. Non-toxic epitopes (toxicity score < 0.5) were retained for vaccine design after rigorous screening through this SVM-based prediction system.

#### 2.1.8. T Cell Epitope Prediction

Cytotoxic T lymphocyte (CTL) and helper T lymphocyte (HTL) epitopes were systematically identified using the IEDB v3.3 (URL: https://www.iedb.org; accessed on 18 April 2024) Analysis Resource. For CTL prediction, artificial neural networks (ANN) were implemented with 8–11 mer peptides and a percentile rank threshold < 1.0 against predominant HLA class I alleles (*HLA-A02:01*, *HLA-B07:02*, *HLA-B35:01*). HTL epitopes were predicted through combinatorial peptide: MHC class II binding affinity matrices, considering 15 mer peptides and HLA-DR allelic variants (*DRB101:01*, *DRB104:01*, *DRB107:01*) [31].

#### 2.1.9. B Cell Epitope Characterization

Linear B cell epitopes (LBEs) were predicted using ABCpred’s RNN v2.1 (URL: https://webs.iiitd.edu.in/raghava/abcpred/; accessed on 18 April 2024) bidirectional recurrent neural network (RNN) model with 16 mer sliding window analysis (threshold: 0.7 specificity). Conformational B cell epitopes (CBEs) were mapped via ElliPro’s Thornton-antibody approach, which combines solvent accessibility and protrusion index calculations. Spatial epitopes were visualized using PyMOL 2.5.8 (accessed on 18 April 2024) with a minimum protrusion index score of 0.6.

#### 2.1.10. Multi-Epitope Vaccine Construction

The multi-epitope vaccine (MEV) was meticulously designed through a hierarchical assembly of immunogenic components. CTL epitopes were interconnected using β-sheet stabilizing AAY spacers. HTL epitopes were joined via GPGPG β-turn promoting linkers, while linear and conformational B cell epitopes (LBEs/CBEs) were bridged by KK dipeptide conjugators [18,32]. For Fc receptor targeting, the human IgG1 Fc domain (UniProt P01857) underwent sequential engineering. First, C226S/C229S substitutions were introduced to eliminate interchain disulfide bonds, ensuring monomerization. Second, *E318A/K320A/K322A* mutations were made to abrogate complement C1q binding. Third, five pH-sensitive residues (*M252Y/S254T/T256E/H433K/N434F*) were modified according to Efgartigimod’s FcRn affinity optimization paradigm to enhance endosomal recycling. Finally, the GGGS flexible connector was utilized to fuse the engineered humanized tFc (h-tFc) into the MEV, ensuring proper spatial orientation between the antigen module and the Fc effector domain.

#### 2.1.11. Solubility Profiling

Protein-SOL’s (URL: https://protein-sol.manchester.ac.uk; accessed on 1 May 2024) machine learning algorithm was employed to predict h-tFc-MEV solubility in E. coli expression systems. The model integrates sequence-derived parameters including charge-hydrophobicity ratio (Z-score < −0.5), aggregation propensity (TANGO score < 5%), and disorder probability (PONDR < 0.4). A normalized solubility score > 0.45 (scale: 0–1) was considered suitable for soluble expression.

#### 2.1.12. Immunostimulation Simulation

Computational immunostimulation simulations were performed using C-ImmSim v10.3.0 (URL: github.com/C-ImmSim; accessed on 1 May 2024), implementing a three-dimensional agent-based model with physiologically scaled lymph node compartments. The vaccine construct was administered in a three-dose regimen (0/4/8 weeks) through FASTA-formatted input, with temporal resolution set at 8 h per simulation unit. HLA restriction elements were selected based on Xinjiang population haplotype frequencies: *HLA-A11:01* (13.46%), *HLA-A02:01* (12.50%), *HLA-A03:01* (10.10%), *HLA-DRB107:01* (16.35%), *HLA-DRB115:01* (8.65%), and *HLA-DRB103:01* (7.69%). Immune parameters included 1000 simulated host equivalents, 200 steps per injection cycle, and germinal center reaction tracking with 85% confidence intervals.

#### 2.1.13. Structural Prediction and Modeling

Protein secondary structure elements were quantified using SOPMA’s (URL: https://npsa-prabi.ibcp.fr/cgi-bin/npsa_automat.pl?page=npsa_sopma.html; accessed on 2 May 2024) self-optimized prediction method with sliding window analysis (window = 17; similarity threshold = 8). Tertiary structures were generated via AlphaFold2 v2.3.0 (accessed on 2 May 2024) multimer mode, employing five recycles with AMBER force field refinement. All models underwent 10 ns molecular dynamics equilibration in explicit solvent.

#### 2.1.14. Structure Refinement and Validation

Initial AlphaFold2 models were optimized using GalaxyWEB 2024Q2 ‘s (URL: galaxyweb.com; accessed on 7 May 2024) loop modeling protocol (ModLoop algorithm) combined with SCWRL4 side-chain repacking. Structural quality was validated through: ERRAT overall quality scores (>85% acceptable regions); Verify3D compatibility (>80% residues scoring > 0.2); PROCHECK Ramachandran analysis (<2% outliers in disallowed regions); and MolProbity clash scores (<10 clashes/100 residues). Final models demonstrated QMEANDisCo global scores > 0.7 and local Z-scores within ±2.0 across all domains.

#### 2.1.15. Molecular Docking and Dynamics Analysis

Molecular docking studies were conducted using the HDOCK v2.0 server (URL: http://hdock.phys.hust.edu.cn/; accessed on 11 May 2024) to evaluate the interaction between the engineered h-tFc-MEV vaccine and human FcRn (PDB ID: 1EXU). Binding metrics included interface ΔG scores calculated using the PISA algorithm, hydrogen bonding networks, and solvent-accessible surface analysis visualized in PyMOL v2.5.0 (accessed on 12 May 2024).

Gromacs2022.3 software (accessed on 15 May 2024) was utilized for molecular dynamics simulation. For small molecule preprocessing, AmberTools was employed to apply the GAFF force field to small molecules, while Gaussian 16W was used for hydrogenation and calculation of RESP potential. The potential data were incorporated into the topology file of the molecular dynamics system. The simulation was conducted at a static temperature of 300 K and atmospheric pressure (1 bar). The Amber99sb-ildn force field was applied, with water molecules serving as the solvent (Tip3p water model), and the total charge of the simulation system was neutralized by adding an appropriate number of Na+ ions. The steepest descent method was utilized to minimize energy, followed by isothermal-isovolumetric (NVT) and isothermal-isobaric (NPT) ensemble equilibrations for 100,000 steps each, with a coupling constant of 0.1 ps and a duration of 100 ps. Finally, a free molecular dynamics simulation was performed, consisting of 5,000,000 steps with a step length of 2 fs, totaling 100 ns. After the simulation, the built-in analysis tool of the software was employed to evaluate the trajectory, calculating the root-mean-square deviation (RMSD), root-mean-square fluctuation (RMSF), and protein rotation radius for each amino acid trajectory, in conjunction with free energy calculations (MMGBSA) and free energy topography [33].

### 2.2. In Vivo and In Vitro Experiment

#### 2.2.1. Construction of Recombinant Plasmid

The MEV and h-tFc-MEV recombinant plasmids were constructed by Yukang Biotechnology through restriction cloning: using *pET-28a* plasmid as template to PCR amplify MEV and h-tFc-MEV genes and integrating the fragments containing MEV and h-tFc-MEV genes into the *pET-19b* plasmid vector via the restriction enzyme sites NdeI and XhoI.

#### 2.2.2. Protein Expression and Purification

The target gene was cloned into the pET-19b vector and subsequently transformed into competent *E. coli BL21(DE3)* cells. Protein expression was induced by the addition of 0.5 mM IPTG (Sigma-Aldrich, I6758; Merck KGaA, Darmstadt, Germany) at a temperature of 18 °C for a duration of 20 h. Bacterial pellets were resuspended in lysis buffer composed of 20 mM Tris-HCl (pH 8.0) (Sigma-Aldrich T6066; Merck KGaA, St. Louis, MO, USA), 300 mM NaCl (Absin, 47052148; Absin Bioscience Inc., Shanghai, China), and 10 mM imidazole (Absin I5513; Absin Bioscience Inc., Shanghai, China), supplemented with a protease inhibitor cocktail (Solarbio P8340; Solarbio Life Sciences Co., Beijing, China), and subsequently lysed using ultrasonication. The lysate was then subjected to centrifugation at 12,000× *g* for 30 min, after which the supernatant was loaded onto a Ni-NTA affinity column (Cytiva 17524802; Cytiva (formerly GE Healthcare Life Sciences), Marlborough, MA, USA). After washing with a buffer containing 50 mM imidazole, the supernatant was subjected to elution under native conditions using a linear imidazole gradient (50 mM to 300 mM). The target protein was detected at 150 mM. It was subsequently desalted and stored at −80 °C.

#### 2.2.3. Western Blot Analysis

Protein concentration was quantified using a BCA assay kit (Solarbio PC0001, Beijing, China). Samples were mixed with SDS loading buffer (Solarbio P1016, Beijing, China), denatured at 95 °C for 8 min, and separated on 12% SDS-PAGE gels (Bio-Rad Laboratories 4568093, Hercules, CA, USA). Proteins were transferred to PVDF membranes (Merck Millipore IPVH00010, Darmstadt, Germany) at 50 V for 60 min. Membranes were blocked with 5% non-fat milk (BD Difco™ 232100, Franklin Lakes, NJ, USA) for 2 h, incubated overnight at 4 °C with anti-His tag rabbit monoclonal antibody (STARTER Biotechnology S0B0006, Wuhan, China; 1:1000 dilution), followed by HRP-conjugated goat anti-rabbit IgG (0RIGENE Biotech ZB-2301, Nanjing, China, 1:5000 dilution) for 1 h at room temperature. After TBST (Servicebio Technology, Wuhan, China) washes, protein bands were visualized using an ECL substrate (Cytiva, Marlborough MSDS-ECL2025, MA, USA, Amersham Imager 600 system).

#### 2.2.4. Preparation and Measurement of Chitosan Nanoparticles

Nanoparticles of MEV and h-tFc-MEV were fabricated employing a chitosan-based ionotropic gelation technique. Briefly, chitosan (1 mg/mL, Heng xing, Zhenjiang City, Jiangsu Province, China) was dissolved in 1% acetic acid (LINSHIHUAXUE 64-19-7, Linyi City, Shandong Province, China) and subsequently filtered through a 0.45 μm membrane. Under magnetic stirring, 5 mL of the chitosan solution (pH adjusted to 4.6) wer mixed with 1.25 mL of recombinant MEV and h-tFc-MEV protein solution at the same concentration (1 mg/mL). Thereafter, a controlled addition of 1.25 mL of sodium tripolyphosphate (1 mg/mL, Heng xing, Zhenjiang City, Jiangsu Province, China) was performed under continuous stirring at room temperature, ensuring thorough gelation over a 1 h period. The resultant mixture was then centrifuged at 10,000× *g* for 30 min to isolate the nanoparticles, followed by careful washing and recovery, adhering to a rigorous protocol to ensure the purity and integrity of the synthesized nanoparticles, suitable for further scientific investigation.

#### 2.2.5. Protein Transport In Vivo

To investigate in vivo transport dynamics, a precise dosage of 70 µg of biotinylated fusion protein was diluted in 200 µL of phosphate-buffered saline (PBS, Servicebio PBS-001, Wuhan, Hubei Province, China). Following oral delivery, the levels of serum transport proteins were meticulously quantified utilizing a highly sensitive enzyme-linked immunosorbent assay (ELISA) kit, (R&D Systems DYC1234-2, Minneapolis, MN, USA) at an 8 h post-administration timepoint [34].

The physicochemical characteristics of the synthesized nanoparticles were meticulously analyzed. The prepared samples were uniformly dispersed in PBS, then diluted for subsequent detection. Dynamic light scattering (DLS, Malvern Instruments, Zetasizer ZS90, Worcestershire, UK) precisely measured the particle size, PDI, and zeta potential of the nanoparticles at different time points (0 h, 8 h, 1 d, 3 d, 5 d, 7 d) and temperatures (4 °C and 25 °C) to ensure a comprehensive understanding of colloidal stability. Particle size and zeta potential were precisely determined via dynamic light scattering, ensuring comprehensive understanding of colloidal stability. Morphological evaluation was undertaken using a high-resolution transmission electron microscope (JEOL JEM-F20, Tokyo, Japan), facilitating visualization of nanostructures.

The total protein amount of initial MEV and h-tFc-MEV was quantified using the BCA Protein Assay Kit (Sigma-Aldrich BCA1, Darmstadt, Germany) assay based on a pre-established BSA standard curve system. Following chitosan encapsulation, samples underwent three consecutive centrifugation cycles (10,000× *g*, 30 min each) to achieve solid-liquid separation. The supernatant generated during centrifugation was completely removed, while the washing solutions from all three cycles were combined and collected. Finally, the free protein amount of unencapsulated MEV and h-tFc-MEV in the pooled supernatant was precisely quantified using the BCA Protein Assay Kit, enabling evaluation of the encapsulation efficiency in the drug delivery system. Encapsulation efficiency was calculated employing the following formula:$$\mathrm{EE}\%=(\frac{Total\;Protein\;Amount-Free\;Protein\;Amount}{Total\;Protein\;Amount})\times 100$$

Subsequently, the nanoparticles were resuspended in PBS to a final concentration of 350 μg/mL for oral immunization, optimizing delivery for subsequent biological evaluation.

#### 2.2.6. Animal Immunization

Female BALB/c mice (6–8 weeks of age, 18–22 g body weight) were randomly divided into three groups and given 200 μL chitosan nanoparticles loaded with h-tFc-MEV (70 μg/dose), MEV protein (70 μg/dose), or PBS control on days 0, 14, and 28, respectively. At the specified endpoint (day 42:14 days after final immunization; day 180: Durability assessment at 6 months), mice were euthanized by CO_2_ asphyxiation, followed immediately by spleen, mesenteric lymph nodes, and enteric laminae propria and serum collection. The study was approved by the Animal Care and Use Committee (ACUC) of XinJiang Medical university, protocol number Agreement K202409-17.

#### 2.2.7. Enzyme-Linked Immunosorbent Assay (ELISA)

##### Sample Collection and Preparation

Serum: Blood was collected from BALB/c mice via retro-orbital plexus puncture under isoflurane anesthesia. After clotting for 30 min at room temperature, serum was separated by centrifugation at 2000× *g* for 15 min (Eppendorf 5804R, Eppendorf AG, Hamburg, Germany). Hemolyzed samples were discarded.

##### Mucosal Lavage Fluids

Bronchoalveolar lavage fluid (BALF): Lungs were lavaged thrice with 1 mL sterile PBS. The pooled fluid was centrifuged at 800× *g* for 10 min (4 °C) to remove cellular debris. Intestinal lavage fluid: Small intestinal lumen was flushed with 5 mL ice-cold PBS. The effluent was centrifuged at 3000× *g* for 15 min to pellet mucus. Vaginal/nasal lavage fluids: Fluids were collected using 1 mL PBS, centrifuged at 2000× *g* for 10 min, and filtered through 0.22 μm membranes (Millipore, Billerica, MA, USA). All samples were aliquoted and stored at −80 °C. Pre-dilutions were optimized as follows: BALF (1:5–1:20), vaginal lavage (1:2–1:10).

High-binding 96-well plates (Costar 3590, Corning, NY, USA) were coated with 50 μL/well of h-tFc-MEV or MEV recombinant protein (2.5 μg/mL in PBS) and incubated overnight at 4 °C. After three washes with PBS containing 0.05% Tween-20 (PBST), plates were blocked with 5% (*w*/*v*) skimmed milk in PBST for 1 h at room temperature. Serially diluted serum samples (twofold dilutions in blocking buffer) were added and incubated for 2 h at 25 °C. Following PBST washes, plates were incubated for 1 h with horseradish peroxidase (HRP)-conjugated antibodies: goat anti-mouse IgA (BioLegend 407002, San Diego, CA, USA; 1:5000), IgG (BioLegend 405306, San Diego, CA, USA; 1:1000), IgG2a (SouthernBiotech 1081-05; Birmingham, AL, USA; 1:6000), or IgG1 (SouthernBiotech 1071-05, Birmingham, AL, USA; 1:8000). After 45 min substrate incubation (37 °C, dark), reactions were developed with 3,3′,5,5′-tetramethylbenzidine (BioLegend 421101, San Diego, CA, USA) for 10 min and terminated with 2 M H2SO4 (Sigma-Aldrich 258105, St. Louis, MO, USA). Absorbance was measured at 450 nm (reference 630 nm) using a SpectraMax M5 microplate reader (Sigma-Aldrich, 258105; St. Louis, MO, USA). Endpoint antibody titers were defined as the highest serum dilution yielding an optical density (OD450) ≥ 0.5, as previously described [35]. Serum IL-21 concentrations were determined using a commercial mouse IL-21 ELISA kit (Sigma-Aldrich 258105, St. Louis, MO, USA) according to the manufacturer’s protocol.

#### 2.2.8. Flow Cytometry

At two weeks post-final immunization, splenic single-cell suspensions were prepared through gentle mechanical dissociation. Cells were stimulated with 5 μg/mL h-tFc-MEV or MEV recombinant protein for 16 h at 37 °C in a 5% CO₂ atmosphere, followed by a 4–6 h incubation with GolgiPlug (BD Biosciences 555029, Franklin Lakes, NJ, USA) to inhibit protein transport. Prior to staining, Fc receptors were blocked using anti-CD16/CD32 antibodies (BD Biosciences 553142, Franklin Lakes, NJ, USA). Surface markers were labeled with fluorochrome-conjugated antibodies, followed by fixation and permeabilization with Cytofix/Cytoperm™ solution (BD Biosciences 554714, Franklin Lakes, NJ, USA) for intracellular staining with anti-mouse IFN-γ and IL-4 antibodies. The washed cells were resuspended in FACS buffer (PBS containing 1% FBS [Gibco 26140079, Grand Island, New York, USA] and 0.1% sodium azide [Sigma-Aldrich S2002, Darmstadt, Germany]) and analyzed by flow cytometry (BD LSRFortessa™, BD Biosciences, Franklin Lakes, NJ, USA). Concurrently, intestinal lamina propria cells from colonic and small intestinal tissues were isolated via enzymatic digestion, while splenocytes were mechanically dissociated. Dendritic cell (DC) subsets were phenotyped using specific fluorescent-conjugated antibodies (CD11c^+^, MHC class II^+^). At the six-month post-immunization time point, lymph node and splenic single-cell suspensions were prepared through gentle mechanical dissociation. Follicular helper T (TFH) cells were identified by CXCR5^+^PD-1^+^ staining, germinal center B (GCB) cells by GL7^+^FAS^+^ markers, and central memory T (TCM) cells by CD44^+^CD62L^+^ expression. Cells were incubated with antibody cocktails for 20 min at 37 °C, protected from light. Flow cytometry data were acquired on a BD FACSLyric system using standardized voltage settings and compensation matrices validated with BD^®^ CompBeads (BD Biosciences 552843, Franklin Lakes, NJ, USA). Data analysis was performed using FlowJo v10.8 software (BD Life Sciences, Ashland, OR, USA), employing pre-established gating strategies that were consistently applied across all experimental time points to ensure analytical continuity.
The Protocol of Flow CytometryIFN-γCD45-APC-cy7 (BD Biosciences 557659, Franklin Lakes, NJ, USA); CD3-APC (Biolegend 100236, San Diego, CA, USA); CD4-FITC (Biolegend 100406, San Diego, CA, USA); CD8-APC-Cy7 (BD Biosciences 557654, Franklin Lakes, NJ, USA); IFN-γ-V500 (BD Biosciences 561980, Franklin Lakes, NJ, USA)IL-4CD45-APC-cy7 (BD Biosciences 557659, Franklin Lakes, NJ, USA); CD3-APC (Biolegend 100236, San Diego, CA, USA); CD4-FITC (Biolegend 100406, San Diego, CA, USA); CD8-APC-Cy7 (BD Biosciences 557654, Franklin Lakes, NJ, USA); IL-4-V450 (BD Biosciences 560701, Franklin Lakes, NJ, USA)DCLive/dead-Bv421 (Thermo Fisher Scientific L34955, Waltham, MA, USA); CD45-APC-cy7 (BD Biosciences 557659, Franklin Lakes, NJ, USA); Lin-APC (BioLegend 133306, San Diego, CA, USA); CD11c-PE-cy7 (BD Biosciences 558079, Franklin Lakes, NJ, USA); MHC-II-FITC (Biolegend 116406, San Diego, CA, USA)TCMCD3-APC (Biolegend 100236, San Diego, CA, USA); CD4-FITC (Biolegend 100406, San Diego, CA, USA); CD8-APC-Cy7 (BD Biosciences 557654, Franklin Lakes, NJ, USA); CD44-PerCP-Cy5.5 (BD Biosciences 1278947, Franklin Lakes, NJ, USA); CD62L-PE-Cy7 (BD Biosciences 1175634, Franklin Lakes, NJ, USA)TFHCD45-APC-cy7 (BD Biosciences 557659, Franklin Lakes, NJ, USA); CD3-APC (Biolegend 100236, San Diego, CA, USA); CD4-FITC (Biolegend 100406, San Diego, CA, USA); CXCR5-PE (BD Biosciences 1165377, Franklin Lakes, NJ, USA); PD-1-Bv605 (BD Biosciences 748267, Franklin Lakes, NJ, USA)GCBCD45-APC-cy7 (BD Biosciences 557659, Franklin Lakes, NJ, USA); CD3-APC (Biolegend 100236, San Diego, CA, USA); B220-FITC (Biolegend 103205, San Diego, CA, USA); GL7-PE (Biolegend 144607, San Diego, CA, USA); FAS-Bv711 (BD Biosciences 740716, Franklin Lakes, NJ, USA)

#### 2.2.9. Immunofluorescence Staining

Bone marrow-derived DCs were stimulated with 5 μg/mL h-tFc-MEV or MEV recombinant protein for 24 h, followed by seeding onto poly-L-lysine-coated coverslips (Sigma-Aldrich, P4707, St. Louis, MO, USA). For fixation, the cells were treated with 4% paraformaldehyde (PFA; Sigma-Aldrich P6148, St. Louis, MO, USA) at 25 °C for 15 min to preserve surface antigens, or with precooled methanol (−20 °C) for 5 min to fix intracellular antigens. After washing with PBS, intracellular antigen detection required permeabilization using 0.1% Triton X-100 (Sigma-Aldrich T8787, St. Louis, MO, USA) for 10 min. Samples were then blocked with 5% bovine serum albumin (BSA; Solarbio A8020, Beijing, China) for 1 h at 25 °C, followed by incubation with cell membrane markers (Thermo Fisher Scientific C10608, Waltham, MA, USA) and anti-His tag primary antibody (STARTER S0B0006, Wuhan, Hubei, China; 1:200 dilution) for 1 h at 37 °C. Subsequently, species-matched Alexa Fluor 488- (Invitrogen A-11001, Carlsbad, CA, USA, 1:500) or Cy3-conjugated secondary antibodies (Jackson ImmunoResearch, 712-165-153, West Grove, PA, USA; 1:500) were applied under light protection. Imaging was conducted using a Nikon A1R confocal microscope (60× oil immersion), with Z-stack sections captured at 0.5 μm intervals and processed using NIS-Elements AR 5.21.

#### 2.2.10. Evaluation of Protective Immunity

To assess protective efficacy, all experimental groups were challenged orogastrically with 10 log10 colony-forming units (CFU) of *Brucella melitensis* smooth strain 16M (ATCC 23456) on day 42 post-vaccination. Animals were euthanized via CO2 asphyxiation on day 56 (14 days post-challenge), followed by the aseptic collection of hepatic, splenic, pulmonary, and mesenteric lymph node tissues. Tissue homogenates (10% *w*/*v* in sterile PBS) were serially diluted in fivefold gradients and plated on 5% defibrinated sheep blood agar (BD Biosciences 221780, Franklin Lakes, NJ, USA) under microaerophilic conditions (5% CO_2_, 37 °C). After 72 h of incubation, Brucella-specific colonies were enumerated using an automated colony counter. Data were normalized to organ weight and expressed as log10 CFU/g ± SEM (*n* = 6/group).

## 3. Results

### 3.1. Physicochemical Properties and Immunological Characteristics of Target Proteins

We investigated the physicochemical properties of VirB10 (UniProt ID: Q8YDZ0) and Omp25 (UniProt ID: Q45321) through computational analyses. The physicochemical properties were evaluated using ProtParam, while antigenicity and allergenicity were predicted using VaxiJen and AllerTop, respectively. As summarized in Table 1, both VirB10 and Omp25 exhibited structural stability and pronounced hydrophilicity. Notably, these proteins demonstrated significant antigenic potential alongside negative profiles in allergenicity prediction, providing compelling immunological evidence for their viability as vaccine candidates.

### 3.2. Screening of Target Protein Sequences

To ensure the safety and efficacy of candidate antigens, a homology analysis against human proteins (*Homo sapiens*, *taxid:9606*) was conducted using the BLAST program for Omp25 and VirB10. The results revealed no significant homology between these antigens and human proteins, thereby substantially mitigating potential immunogenicity risks. During the optimization of antigen design, signal peptide characteristics were predicted using SignalP 5.0. Omp25 displayed a Sec/SPI-type signal peptide cleavage site between amino acid residues 23 and 24 (AFA-AD motif) with a confidence score of 0.9303, whereas VirB10 did not exhibit any detectable signal peptide domain. Given the critical biological role of signal peptides in directing nascent proteins to the endoplasmic reticulum [36], the 1–23 amino acid signal peptide sequence of Omp25 was ultimately removed during recombinant expression to ensure proper localization and functional integrity in host cells (Appendix A).

To optimize vaccine targeting, this study focused on high-frequency HLA alleles prevalent in the Xinjiang region, specifically *HLA-A11:01* (13.46%), *HLA-A02:01* (12.50%), *HLA-A03:01* (10.10%), *HLA-DRB107:01* (16.35%), *HLA-DRB115:01* (8.65%), and *HLA-DRB103:01* (7.69%). Epitope prediction was conducted using the IEDB database, selecting 10-mer peptides for CTL epitopes (MHC-I binding) and 15-mer peptides for HTL epitopes (MHC-II binding). Binding affinity scores between epitopes and MHC molecules were calculated based on the amino acid sequences of Omp25 and VirB10. The top 10 high-affinity epitopes from each target protein were prioritized for subsequent immunogenicity analysis to identify candidates with robust immune activation potential.

To systematically evaluate the B-cell epitope profiles of Omp25 and VirB10, a multi-strategy prediction approach was employed: (1) the ABCpred tool was utilized to screen 16-mer linear epitopes, emphasizing their antigen-binding potential, and (2) structural bioinformatics algorithms on the IEDB platform were applied to analyze three-dimensional structural data of Omp25 (PDB ID: Q45321) and VirB10 (PDB ID: Q8YDZ0) from the UniProt database for conformational epitope prediction. This dual analytical strategy aimed to comprehensively elucidate the B-cell epitope landscape of the candidate antigens, providing precise immunological targets for subsequent vaccine design.

Ensuring stability, biocompatibility, and safety is paramount in vaccine development. Vaccines must maintain structural stability under variable storage and transportation conditions [37,38], preventing degradation to preserve immunogenic efficacy. Optimal hydrophilicity and solubility are critical for rapid in vivo distribution and enhanced immune response amplification [39,40]. The core design challenge lies in selecting antigenic peptides capable of eliciting robust immune responses while ensuring that these epitopes avoid allergenic reactions that might interfere with immunological mechanisms [41]. Therefore, ideal vaccine epitopes should exhibit high stability, hydrophilicity, strong antigenicity, and guaranteed non-toxicity with non-allergenic properties. Specific selection criteria include an antigenicity score greater than 0.4 for protective efficacy, a hydrophilicity index of less than 0, a stability score of less than 40, and rigorous evaluation to exclude toxicity and sensitization risks. Based on these stringent parameters, Appendix A summarizes the screening outcomes, identifying six potential CTL epitopes, five HTL epitopes, one CBE epitope, and eight LBE epitopes, thus establishing a robust foundation for vaccine development.

### 3.3. Vaccine Design and Structural Assembly

The chimeric h-tFc-MEV vaccine was engineered through the systematic integration of immunodominant epitopes and modifications to the Fc domain. Computational screening identified six CTL epitopes, five HTL epitopes, eight LBE epitopes, and one CBE based on criteria including antigenicity, hydrophilicity, non-toxicity, and non-allergenicity. These epitopes were assembled into an MEV using protease-cleavable linkers: CTL epitopes were flanked by AAY spacers to enhance MHC-I presentation, HTL epitopes were interconnected by GPGPG motifs for MHC-II processing, and LBEs were conjugated to CBEs via KK linkers to preserve conformational integrity (Figure 1D,E). The structure of the FcRn receptor is based on the combination of its α chain and β2 microglobulin, which specifically recognizes the Fc region of IgG, especially the CH2-CH3 domain, thanks to the interaction of specific amino acid residues [42,43] (Figure 1C). The FcRn-targeting backbone was derived from human IgG1 Fc (UniProt ID: P01857), selected for its high affinity to FcγRI and low binding to FcγRIIB [28]. Key modifications included the following: C1q-binding ablation via *E318A/K320A/K322A* mutations to eliminate complement activation; monomerization through *C226S/C229S* substitutions, disrupting interchain disulfide bonds to prevent dimerization; and FcRn affinity enhancement via Efgartigimod-inspired mutations (*M252Y/S254T/T256E/H433K/N434F*), engineered using ABDEG technology to strengthen pH-dependent FcRn binding, thereby improving stability across physiological pH gradients and extending serum half-life [44,45]. The optimized h-tFc module (Appendix A) was fused to the MEV via a flexible GGGS linker, forming the h-tFc-MEV complex. This design leverages FcRn-mediated transcytosis for efficient mucosal delivery while maintaining epitope immunogenicity (Figure 1A,B).

### 3.4. Physicochemical Properties and Immunological Characteristics

Computational analysis of MEV and h-tFc-MEV revealed favorable physicochemical profiles, including high antigenicity, non-allergenicity, stability, and hydrophilicity. Additional properties are summarized in Table 2.

### 3.5. Immune Simulation

Immunoinformatics simulations revealed that h-tFc-MEV vaccination elicited a balanced immune activation without excessive modulation of DCs. The counts of DCs stabilized at 180 cells/mm^3^, with 20 cells/mm^3^ retaining active phenotypes (Figure 2A), indicating neither hyperactivation nor suppression of DC functionality. T-cell responses demonstrated progressive amplification: TH cell populations increased incrementally across three immunizations (Figure 2B), with activated and resting CD4+ T cells peaking at 6200 cells/mm^3^ and 4100 cells/mm^3^, respectively (Figure 2C). B-cell responses peaked following the third immunization, achieving total and activated counts of 760 cells/mm^3^ (Figure 2D,E), consistent with robust humoral immunity. In *Brucella* infection models, IgG1 predominates in pathogen clearance by targeting surface antigens, while IgM mediates early neutralization [21]. h-tFc-MEV immunization induced IgG titers that peaked at 380,000, confirming a Th1-skewed immune response, alongside a synchronized elevation of IgM (peak: 400,000) (Figure 2F), demonstrating a coordinated, rapid, and sustained defense. In the context of *Brucella* infection, IFN-γ enhances bactericidal capacity by activating macrophages, while IL-2 promotes T-cell proliferation to strengthen the immune response [46]. Conversely, overexpression of IL-4 may disrupt the Th1/Th2 balance and weaken host defense [47]. Following three doses of the h-tFc-MEV vaccine, levels of IFN-γ, IL-2, IL-4, and antibodies were significantly elevated (Figure 2G). Immunological simulations indicated that the vaccine could activate dendritic cells, drive the expansion of CD4+ T cells and B cells, and induce Th1-dominant immunity (primarily IgG1) along with multicellular secretion, confirming its potential as a candidate for an anti-*Brucella* vaccine.

### 3.6. Secondary and Tertiary Structure Prediction of Vaccines

The secondary structure of h-tFc-MEV was predicted using the SOPMA algorithm, revealing a composition of 20.53% α-helices, 54.06% random coils, and 25.41% extended strands (Figure 3A). Subsequent tertiary structure prediction was conducted using AlphaFold2 (AF2), which identified the highest-confidence model (pLDDT = 56, pTM = 0.347) that underwent structural refinement. The optimized model was visualized in three dimensions utilizing Discover Studio 2023 software (Figure 3B). Notably, pLDDT (a per-residue confidence score ranging from 0 to 100) and pTM (predicted TM score) collectively indicate the reliability of the model. The distribution patterns of hydrogen bonds were further illustrated (Figure 3C). Importantly, the tertiary structural composition (α-helices: 20.51%, random coils: 54.08%, extended strands: 25.41%) exhibited remarkable consistency with secondary structure predictions, showing less than 0.5% variation. This demonstrates a strong concordance between AF2 modeling and experimental structural analyses, thereby validating the accuracy of the tertiary structure prediction.

### 3.7. Molecular Interaction Mechanism of h-tFc-MEV/FcRn Complex

To investigate the molecular interaction mechanism between h-tFc-MEV and FcRn, molecular docking simulations were performed using AlphaFold v2.3.2, followed by three-dimensional visualization of the binding interface in PyMOL v2.5.0, which revealed strong binding affinity (Appendix A). Molecular dynamics simulations and multidimensional analyses demonstrated enhanced structural and thermodynamic stability of the h-tFc-MEV/FcRn complex. RMSD analysis showed system equilibration after 15 ns with fluctuations stabilized at 0.1–0.3 nm, where the h-tFc-MEV system exhibited lower conformational flexibility than the unmodified counterpart. Structural compaction was evidenced by reduced radius of gyration (Rg = 3.65 nm, 17% decrease from initial state) and lower solvent-accessible surface area (SASA = 600 nm^2^), indicating effective burial of hydrophobic cores. The interface maintained 28–33 hydrogen bonds on average, with prolonged persistence in the modified system. RMSF analysis revealed restricted fluctuations (<2.0 Å) at binding residues, while non-binding loops showed higher mobility, though residues near h-tFc-MEV modification sites displayed enhanced dynamic stability. Free energy landscape analysis identified a concentrated low-energy basin (energy gap <10 kJ/mol) for h-tFc-MEV, contrasting with the dispersed energy states of the unmodified system, suggesting reduced conformational entropy. MM/PBSA calculations quantified the binding free energy as ΔGbind = −66.91 kJ/mol, significantly exceeding the threshold for strong interactions. These findings collectively demonstrate that h-tFc-MEV strengthens FcRn binding through reinforced hydrogen bonding, optimized structural rigidity, and thermodynamic stabilization, providing a molecular basis for its extended plasma half-life (Appendix A).

### 3.8. Western Blot Analysis of MEV and h-tFc-MEV

Based on the predicted stability and dual-modal immune activation properties of h-tFc-MEV, we evaluated its biological functions through in vitro and in vivo experiments. The recombinant MEV and h-tFc-MEV proteins were expressed in E. coli pLysS using the *pET-19b* plasmid system. Following induction with 0.5 mM IPTG, proteins were purified under native conditions via Ni-NTA affinity chromatography with a linear imidazole gradient (elution at 150 mM). Western blot analysis using an anti-His tag antibody confirmed the purity of both proteins, revealing distinct bands at 46 kDa (MEV) and 71.54 kDa (h-tFc-MEV) that closely matched their theoretical molecular weights calculated from amino acid sequences (Appendix A). These results demonstrate the structural integrity and accurate post-translational processing of the recombinant proteins.

### 3.9. Engineering a Chitosan-Based Nanoparticle System for Enhanced Mucosal Vaccine Delivery

To overcome the biological barriers of macromolecular drug delivery including low mucosal permeability, inadequate lipophilicity, and enzymatic degradation in the gastrointestinal tract, we developed a chitosan (CS)-based nanoparticle system for efficient antigen delivery. Comprehensive characterization using dynamic light scattering (DLS) and transmission electron microscopy (TEM) revealed critical advantages of the engineered h-tFc-MEV-CS and MEV-CS nanoparticles:

First, both types of nanoparticles exhibited a regular spherical morphology. Over time, although the particle size increased, storage at 4 °C significantly improved stability compared to room temperature (25 °C) (Figure 4A(a,d)). Notably, in the early stage after preparation (0–8 h), the average particle size of chitosan nanoparticles could be controlled within 200 nm (Figure 4B(a,b,d,e)), meeting the basic size requirements (<200 nm) for effective penetration of intestinal mucus [48,49] and laying a good physical foundation for subsequent delivery processes.

Second, in terms of polydispersity index (PdI), the trend of PdI changes over time also indicated that the PdI fluctuation of nanoparticles stored at 4 °C was much smaller than that at room temperature (25 °C), highlighting the significant stability advantage. In the early stage after preparation (0–8 h), the PdI of chitosan nanoparticles could stably remain below 0.3, which is a key piece of evidence of their ideal nanoparticle vaccine characteristics, implying that they can maintain a relatively uniform particle distribution during delivery, thus facilitating the smooth activation of subsequent immune responses.

Third, focusing on the charge characteristics, the surface charge intensity of the nanoparticles remained relatively stable with only a slight decrease over a seven-day observation period, and the overall surface charge was positive. Moreover, storage at 4 °C again demonstrated superior stability compared to room temperature (25 °C) (Figure 4A(a–c,f)). Specifically, in the early stage after preparation (0–8 h), the zeta potential of chitosan nanoparticles reached +33.3 ± 1.45 mV and +32.2 ± 1.15 mV (Figure 4B(a–c,f)). This positive charge enables the nanoparticles to easily adhere to the mucosa via electrostatic interaction with negatively charged epithelial surfaces and greatly promotes cellular internalization, creating favorable conditions for effective antigen presentation and potentially inducing a more robust immune response.

In addition to the above key characteristics, the protein loading efficiencies of h-tFc-MEV and MEV were also satisfactory, reaching 68.52 ± 2.31% and 67.36 ± 1.97%, respectively. This ensures that a sufficient amount of antigen can be safely and stably loaded onto the nanoparticles to meet the demands of subsequent immune activation.

In summary, the newly prepared chitosan nanoparticles, with their nanoscale size, excellent surface charge properties, and superior payload capacity, have the potential to serve as effective mucosal vaccine delivery systems.

### 3.10. Enhanced Mucosal Penetration of h-tFc-MEV in Oral Delivery Models

To evaluate the mucosal penetration capacity of the h-tFc-MEV protein, we established an oral delivery model using chitosan-based nanocarriers. Mice were randomly allocated into three experimental groups and administered via oral gavage 70 μg of either chitosan nanoparticle-encapsulated h-tFc-MEV proteins or MEV proteins or PBS. Serum antigen levels were quantified at 8 h post-administration using ultrasensitive ELISA. Quantitative analysis revealed a statistically significant elevation in serum protein concentration in the h-tFc2212MEV group compared to the MEV group (3.63-fold increase, *p* < 0.01; Figure 5). This result demonstrates the superior trans-mucosal transport efficiency of h-tFc-MEV, which may be attributed to its enhanced interaction with mucosal surfaces mediated by the engineered h-tFc domain.

### 3.11. Enhanced Humoral Immunity with Optimized Th1/Th2 Profiling

Serological analysis at 14 days post-final immunization revealed a 1.54-fold increase in total antigen-specific IgG titers in the h-tFc-MEV group compared to the MEV group (*p* < 0.01, Figure 6A). While the IgG2a/IgG1 ratio showed a modest reduction in the h-tFc-MEV group (1.11 vs. 1.23 in MEV group), its absolute value remained significantly above the Th2-dominant response threshold (ratio > 1.0, Figure 6B). This observation, combined with the markedly elevated total antibody levels, suggests that h-tFc-MEV achieves comprehensive enhancement of antibody response magnitude while preserving Th1-skewed immunogenicity. Notably, although the Th1 polarization index (based on IgG subclass ratios) showed a relative decrease, the absolute antibody titers (total IgG) in the h-tFc-MEV group exhibited superior efficacy compared to MEV. These findings imply that the vaccine’s optimized antigen delivery and presentation efficiency may overcome conventional limitations in Th1/Th2 balance regulation, potentially through enhanced dendritic cell uptake and cross-presentation mediated by the engineered h-tFc domain.

### 3.12. FcRn Mediated Subunit Vaccine Strategy Significantly Improve Antigen-Presenting Effect

The FcRn-targeted delivery strategy has emerged as a pivotal approach in subunit vaccine design by enhancing IgG-mediated immune complex recycling within DCs, thereby amplifying both MHC I and MHC II antigen presentation pathways to activate CD8+ and CD4+ T cell responses [50]. In vitro experiments demonstrated that h-tFc-MEV-treated DCs exhibited 2.02-fold higher antigen presentation efficiency compared to MEV-treated counterparts after 24 h incubation (Figure 7A,B), a phenomenon mechanistically linked to FcRn-mediated optimized endocytic trafficking. Subsequent in vivo validation through flow cytometry analysis revealed that h-tFc-MEV immunization induced 2.28-fold, 3.11-fold, and 2.60-fold increases in mature DC populations within spleen and small intestine and large intestine lamina propria, respectively (Figure 7C,D; gating strategy in Appendix A), accompanied by 1.41-fold elevated IL-12 production in splenic tissues (Figure 7E). These coordinated findings substantiate that FcRn-driven vaccine engineering not only potentiates DC antigen-presenting capacity but also systemically activates key effector cells, providing a robust immunological foundation for vaccine efficacy enhancement.

### 3.13. h-tFc-MEV Elicits Pathogen-Specific T-Cell Immunity

The h-tFc-MEV vaccine demonstrated robust enhancement of T-cell-mediated immune protection by mimicking pathogen infection. Flow cytometric analysis (BD FACS Canto II; gating strategy in Appendix A) revealed a Th1-skewed immune profile in splenocytes 14 days post-final immunization: h-tFc-MEV immunization elevated the frequencies of IFN-γ^+^CD4^+^and CD8^+^T cells by 2.52-fold and 2.44-fold, respectively, compared to the MEV group (Figure 8). Notably, concurrent Th2-polarized responses were observed, with IL-4^+^CD4^+^and CD8^+^T cells increasing by 2.65-fold and 5.66-fold, respectively (Figure 8). Strikingly, h-tFc-MEV overcame the limitation of MEV (which showed no significant difference versus PBS in CD8^+^IL-4^+^ T cells), demonstrating its dual modulation of Th1/Th2 immunity. These findings underscore that FcRn-targeted delivery optimizes CD4^+^/CD8^+^ T-cell balance and Th1/Th2 synergy, providing a novel vaccine strategy against intracellular pathogens like *Brucella*.

### 3.14. FcRn-Targeted Strategy Enhances Long-Term Immunological Memory Induced by h-tFc-MEV Vaccine: Pivotal Roles of Central Memory T Cells and Germinal Center Activity

This study demonstrates that FcRn-targeted mucosal immunization significantly enhances antigen-specific TCM cell levels for at least six months post-boost, while maintaining elevated frequencies of GCB and TFH cells—key cellular determinants of potent and sustained immune memory.

TCM cells play a pivotal role in vaccine-induced long-term memory. Upon re-exposure, these cells rapidly activate to amplify immune responses, thereby ensuring robust defense [51]. In h-tFc-MEV-vaccinated mice, sustained expansion of CD4^+^ (7.79 ± 0.91%) and CD8^+^ (40.90 ± 6.22%) TCM subsets persisted for 180 days post-immunization, exhibiting 2.31- and 3.07-fold higher frequencies compared to PBS controls, respectively. When compared to MEV-vaccinated counterparts, these values represented 2.06- and 1.51-fold increases (Figure 9A; gating strategy in Appendix A). This persistence of TCM suggests accelerated recall responses during potential *Brucella* exposure, thereby enhancing protective efficacy.

In an in-depth evaluation of vaccine efficacy, the immune dynamics in the lymphoid organs of mice vaccinated with h-tFc-MEV were systematically monitored over a period of two weeks and six months. The study revealed sustained activation of germinal centers in both lymph nodes and spleens, where TFH and GCB cells established a stable immune synergy through an IL-21-mediated paracrine network. This cellular interaction facilitated B cell clonal expansion, plasma cell differentiation, and the formation of a memory B cell pool, thereby underpinning long-term immunity. Flow cytometry analysis conducted six months post-immunization indicated that the h-tFc-MEV group maintained TFH cell proportions of (18.00 ± 3.17)% in lymph nodes (gating strategy in Appendix A, Figure 9B(a)) and (12.36 ± 2.00)% in spleens (Figure 9B(b)), representing 7.40- and 6.31-fold increases over PBS controls, respectively, and 1.81- and 1.95-fold advantages over MEV controls. Concurrently, GCB cells reached (0.98 ± 0.19)% in lymph nodes (gating strategy in Appendix A, Figure 9C(a)) and (2.70 ± 0.31)% in spleens (Figure 9C(b)), with enhancements of 4.98- and 28.8-fold compared to PBS groups and improvements of 2.03- and 3.05-fold over MEV groups. ELISA confirmed elevated splenic IL-21 levels (545.8 ± 64.3 pg/mL, Figure 9D(a)), which correlated with the expansion of TFH and GCB cells. Despite a decline in antibody titers, serum analysis (Figure 9D(b)) demonstrated that high protective IgG levels were preserved in the h-tFc-MEV group, highlighting its dual strengths in facilitating a rapid early-phase response and establishing durable immune memory.

### 3.15. Mucosal Immune Responses Induced by h-tFc-MEV Vaccine via FcRn-Targeting Strategy

The production of mucosa-specific antibodies in mucosal secretions constitutes a critical component of the mucosal immune response. At two weeks post-vaccination, lavage fluids were collected from the intestinal lumen, bronchoalveolar space, nasal cavity, and vaginal tract for precise quantification of vaccine-specific IgG and IgA levels using ELISA. The results demonstrated that h-tFc-MEV vaccination significantly enhanced both IgA and IgG production in intestinal and pulmonary mucosa compared to the PBS group (*p* < 0.01), indicating robust proximal mucosal immune responses. Notably, h-tFc-MEV induced markedly higher IgA and IgG levels in pulmonary mucosa than MEV (*p* < 0.01), whereas intestinal IgA/IgG elevation showed a weaker statistical significance (*p* < 0.05). In nasal lavage fluid, h-tFc-MEV elicited elevated IgG and IgA compared to PBS (*p* < 0.01), but no significant difference was observed versus MEV (*p* > 0.01). Vaginal secretions exhibited detectable IgG titers, with a moderate increase in h-tFc-MEV over PBS (*p* < 0.05), yet no divergence from MEV. Additionally, h-tFc-MEV significantly boosted IgA in vaginal mucosa relative to PBS (*p* < 0.01), but not compared to MEV (*p* > 0.01). These findings highlight that h-tFc-MEV preferentially activates local mucosal IgA responses in gut-associated and bronchus-associated lymphoid tissues, while providing limited protection in distal mucosal compartments, underscoring the region-specific nature of mucosal immunity (Figure 10).

### 3.16. Protective Efficacy of Orally Administered FcRn-Targeting Vaccine

Animal challenge experiments demonstrated significant mucosal barrier protection and systemic organ defense mediated by h-tFc-MEV vaccination. Mice receiving three-dose prime immunization were orally challenged with 10.3 log10 CFU *Brucella melitensis*, and bacterial loads in tissues were quantified via plate culture at 14 days post-challenge. The h-tFc-MEV group exhibited a 1.7 log10 CFU/g reduction in mesenteric lymph node bacterial burden compared to PBS controls, with a 0.77 ± 0.43 log10 CFU/g advantage over MEV group (*p* < 0.01), indicating vaccine-specific suppression of intestinal pathogen dissemination.

Systemically, h-tFc-MEV vaccination achieved superior protection: splenic bacterial load (2.365 ± 0.62 log10 CFU/g) decreased by 28.2% ± 19.1% versus MEV controls (*p* < 0.01), while hepatic burden (1.62 ± 0.34 log10 CFU/g) showed a 25.7% ± 17.4% reduction (*p* < 0.01). Lung protection displayed tissue specificity, with a 39.5% ± 26.4% decrease relative to PBS (2.76 ± 0.25 log10 CFU/g; *p* < 0.01), but only a 5.7% ± 64.3% reduction compared to MEV (1.77 ± 0.91 log10 CFU/g; *p* > 0.01). Renal protection exhibited stratified efficacy, achieving 72.7% ± 24.5% bacterial clearance versus PBS (2.49 ± 0.37 log10 CFU/g; *p* < 0.01), yet showing a 46.7% ± 75.3% reduction relative to MEV (1.28 ± 0.63 log10 CFU/g; *p* > 0.01) (Figure 11). These data underscore the compartmentalized protective profile of FcRn-targeted mucosal vaccination.

The h-tFc-MEV vaccine elicits dual mucosal-systemic immune responses, effectively blocking *Brucella* invasion through gastrointestinal and respiratory routes while establishing robust protective effects in reticuloendothelial system organs (spleen, liver, and lungs) by suppressing pathogen colonization and replication. These findings provide critical experimental evidence for developing mucosal vaccines against brucellosis.

## 4. Discussion

To address the limitations of immunological barriers in traditional multi-epitope vaccines, this study innovatively introduced an FcRn receptor-targeting mechanism—by enhancing the specific binding between humanized truncated h-tFc and FcRn to promote reverse transcytosis of IgG across intestinal mucosal surfaces, thereby establishing a novel strategy for long-lasting mucosal immunity.

In designing a Brucella multi-epitope vaccine (MEV), we chose VirB10 and Omp25, two key Brucella outer membrane proteins, as candidate antigens. To ensure their proper intracellular localization and function [36], we precisely removed their signal peptide sequences using bioinformatics and experimental validation. For T-cell epitopes, we analyzed their characteristics. These 8–20 amino acid fragments can form complexes with MHC molecules for T-cell receptor recognition. MHC-I presents endogenous epitopes to activate CTLs for infected cell killing; MHC-II presents exogenous ones to activate HTLs and initiate adaptive immune responses [52]. Thus, selecting T-cell epitopes is vital for triggering cellular immunity. We also studied B-cell epitopes. B cells recognize linear (LBEs) or conformational (CBEs) epitopes via BCRs [52]. LBEs are continuous sequences; CBEs rely on protein spatial structures. Selecting B-cell epitopes is crucial for activating B cells to produce specific antibodies. After optimizing antigens and screening epitopes, we used biocompatible linkers to connect key epitopes, forming the MEV prototype. This strategy integrates Brucella immunogenic elements to activate both cellular and humoral immunity, aiming for effective Brucella infection prevention.

The FcRn-targeting approach significantly improves mucosal vaccine efficacy, with FcRn-based vaccines demonstrating favorable immune tolerance profiles [34,53]. Specifically, we pioneered a mucosal-targeted delivery strategy through fusion of the h-tFc domain with a MEV, constructing an innovative oral vaccine h-tFc-MEV against *Brucella*. Through comprehensive bioinformatic evaluation (including antigenicity index > 0.4, hydrophilicity [GRAVY score < 0], structural stability, and non-allergenicity), we identified conserved T/B-cell dominant epitopes from *Brucella* Omp25 outer membrane protein and VirB10 secretion system protein with cross-species conservation. These epitopes were subsequently assembled into the MEV fusion protein using flexible linker sequences.

In structural optimization, the h-Fc was engineered based on the Efgartigimod-derived model to generate a truncated h-tFc variant, which exhibited enhanced binding affinity to FcRn under intestinal acidic conditions (pH 6.0) while maintaining structural stability at physiological pH (7.4). Bioinformatic predictions revealed favorable physicochemical properties of the h-tFc-MEV construct, including strong hydrophilicity (GRAVY score: −0.577), high antigenicity (antigenicity index: 0.6878), non-allergenicity, and structural stability. Immunological simulations demonstrated its capacity to activate dendritic cells and promote CD4+ T/B lymphocyte proliferation, inducing Th1-polarized immune responses with concomitant secretion of multiple cytokines including IFN-γ and IL-4. Molecular docking and MM-PBSA calculations using the AlphaFold2-predicted tertiary structure model showed strong binding affinity between h-tFc-MEV and FcRn (binding free energy: −66.91 kJ/mol), with 100 ns molecular dynamics simulations confirming complex stability. These integrated findings collectively demonstrate the candidate vaccine’s potential to overcome mucosal immune barriers.

To address gastrointestinal mucosal barrier challenges, we innovatively developed a CS-based nanoparticle delivery system. Capitalizing on CS’s inherent properties of high biodegradability, cytocompatibility, and antimicrobial activity [54], the system incorporates surface functionalization via FcRn-targeting ligand modification to achieve precision delivery [55]. The engineered nanoparticles (100–200 nm diameter) not only protect antigens from enzymatic degradation but also enhance intestinal epithelial uptake through improved cellular membrane permeability [56], consistent with the 10-fold delivery efficiency enhancement reported by Pridgen et al. for FcRn-targeted nanoparticles [57]. In our chitosan-based oral delivery model evaluating mucosal penetration capacity, the h-tFc-MEV group exhibited statistically significant elevation in serum protein concentrations compared to MEV controls, demonstrating superior transmucosal transport efficiency that aligns with Pridgen’s findings.

As a pivotal receptor regulating IgG and albumin recycling, FcRn-mediated immune complex (IC) transport mechanisms offer novel avenues for vaccine design. Classical studies established that FcRn not only extends IgG half-life via recycling [58] but crucially orchestrates antigen cross-presentation: within antigen-presenting cells (APCs) like dendritic cells, multivalent IgG ICs initiate specific transport programs through FcRn crosslinking, directing antigens to MHC class I/II processing compartments to coordinately activate CD4+/CD8+ T-cell responses [59]. Notably, antigen uptake mediated solely by FcγR fails to achieve cross-presentation [60], underscoring FcRn’s irreplaceable role in bridging humoral and cellular immunity. This study demonstrates that FcRn-driven antigen transport enhances MHC-II molecule expression on dendritic cells and improves antigen presentation efficiency. Our experimental data further reveal that FcRn-mediated antigen delivery overcomes limitations of conventional vaccine-induced immune response patterns, providing a breakthrough theoretical foundation for developing next-generation vaccines with prolonged protection and broad-spectrum T-cell immunity.

Following three oral immunizations, the experimental group exhibited multidimensional immune responses within a short-term period (14 days post-final vaccination). The FcRn-targeted h-tFc-MEV group demonstrated significantly elevated serum IgG titers compared to MEV controls, accompanied by enhanced secretory IgA levels in key mucosal defense sites (intestinal, respiratory, and reproductive tracts). Concurrent activation of both Th1/Th2 pathways was evidenced by substantial increases in IFN-γ and IL-4 secretion, confirming effective induction of antigen-specific CD4+ T-cell responses. Notably, the observed IL-4+ T-cell population proportions by flow cytometry diverged from those reported by Jia Wen et al. [61], potentially attributable to differential epitope presentation influenced by molecular linkage patterns in our t-Fc segment design, which warrants further experimental validation. This synergistic activation of systemic and mucosal immunity provides mechanistic insights into rapid immune protection. In long-term evaluations (6 months post-immunization), while serum IgG titers in h-tFc-MEV group remained significantly higher than MEV controls, they declined compared to the 14-day peak.

Immunological memory is characterized by the expansion of effector T and B cell pools and rapid, amplified recall responses upon re-exposure to pathogens. A key goal of vaccine design is to establish high-quality memory lymphocyte populations capable of long-term immune surveillance and protection. Central memory T cells (TCM) are central to vaccine-induced long-term memory. After activation, pathogen-specific T cells differentiate into memory states, persisting as immunological sentinels [51]. Notably, high-frequency GCB, TFH and TCM cells have been detected in lymphoid tissues, indicating long-term residency of antigen-specific immunological memory reservoirs and providing strong evidence for durable mucosal vaccine protection.

In challenge models, h-tFc-MEV substantially reduced *Brucella* loads in spleen, liver, mesenteric lymph nodes, and kidneys, demonstrating effective suppression of organ colonization. This multi-organ clearance synergy aligns with previously observed dual-track mucosal-systemic immunity, suggesting that FcRn-mediated antigen delivery not only enhances immune priming but also establishes defensive networks through rapid tissue homing of memory T/B cells.

Despite groundbreaking advancements, this study necessitates refinements in two critical aspects. First, N-glycosylation modifications of intestinal FcRn in non-human primates may induce steric hindrance between chitosan nanoparticles and surface ligands, potentially causing interspecies discrepancies in targeting efficiency. Second, the current MEV epitope library, primarily designed against *B. melitensis*, lacks integration of antigenic epitopes from clinically significant species like *B. abortus*. To address these challenges, subsequent studies will employ directed evolution to screen broad-spectrum Fc variants and combine bioinformatic prediction with phage display technology to construct multi-epitope chimeric antigens covering six clinically significant *Brucella* pathogenic subtypes. By integrating structure-guided antigen design with cross-species targeting strategies, this platform not only establishes a novel paradigm for developing universal mucosal vaccines against intracellular pathogens but also demonstrates translational potential in combating public health threats like multidrug-resistant tuberculosis. Collectively, the FcRn-targeted mucosal vaccine delivery system, through synergistically activating mucosal-systemic immunity, pioneers an innovative intervention strategy for brucellosis prevention and control.

## 5. Conclusions

This study developed a brucellosis mucosal vaccine (h-tFc-MEV) leveraging FcRn-targeting mechanisms through genetic fusion of human Fc domains with MEV antigens to achieve antigen-FcRn synergy. Oral administration induced dual immune activation: serum IgG and mucosal sIgA levels were significantly elevated alongside sustained activation of GCB, TFH, and TCM cells. Challenge experiments demonstrated enhanced dendritic cell antigen presentation efficiency via FcRn-mediated antigen transport, effectively suppressing *Brucella* multi-organ colonization. This modular design platform overcomes the immunogenicity limitations of conventional mucosal vaccines, providing an innovative strategy for combating intracellular pathogens like brucellosis. Its cross-pathogen adaptability holds significant translational potential in addressing public health threats such as multidrug-resistant tuberculosis.

## Figures and Tables

**Figure 1 vaccines-13-00567-f001:**
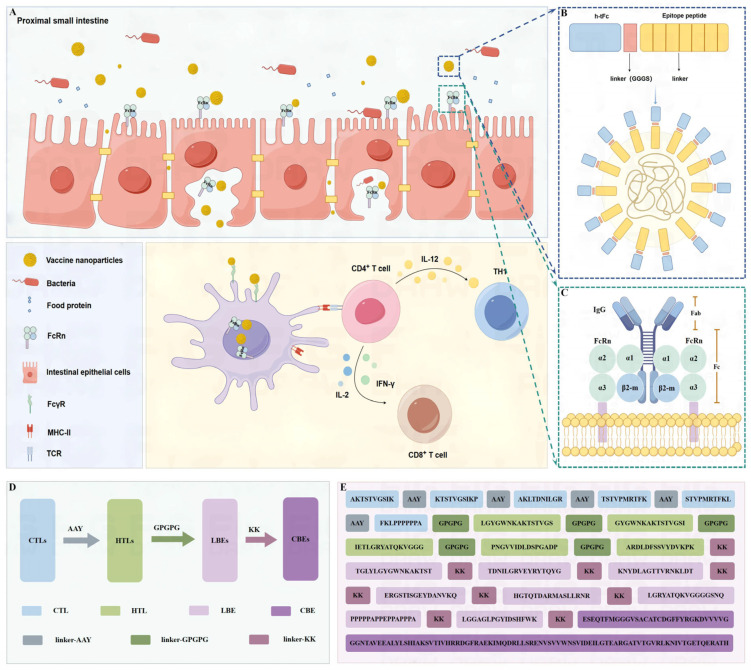
Mechanism and molecular design of FcRn-targeted nanoparticle vaccine. (**A**) Activation of the intestinal mucosal immune response occurs when Fc-conjugated nanoparticles are transcytosed across the intestinal epithelium via FcRn. DCs in the lamina propria capture these nanoparticles through FcγR, process the antigens via MHC-II, and subsequently activate CD4+ T cells. The activated CD4+ T cells secrete IL-21, promoting T cell polarization. Additionally, they release IL-2, which facilitates T cell proliferation, and IFN-γ, which enhances immune responses. This synergistic action promotes DC-mediated cross-presentation through MHC-I, leading to the activation of CD8+ T cells, which differentiate into CTLs that contribute to systemic mucosal immunity. (**B**) The structure of the nanoparticle is depicted in a schematic diagram showing the formation of chitosan nanoparticles for the vaccine. (**C**) An interaction diagram illustrating the FcRn-IgG relationship is provided. (**D**) A schematic representation of the MEV connection is included. (**E**) The sequence of the MEV is presented, comprising T/B cell epitopes and linkers. The image is created using figdraw.

**Figure 2 vaccines-13-00567-f002:**
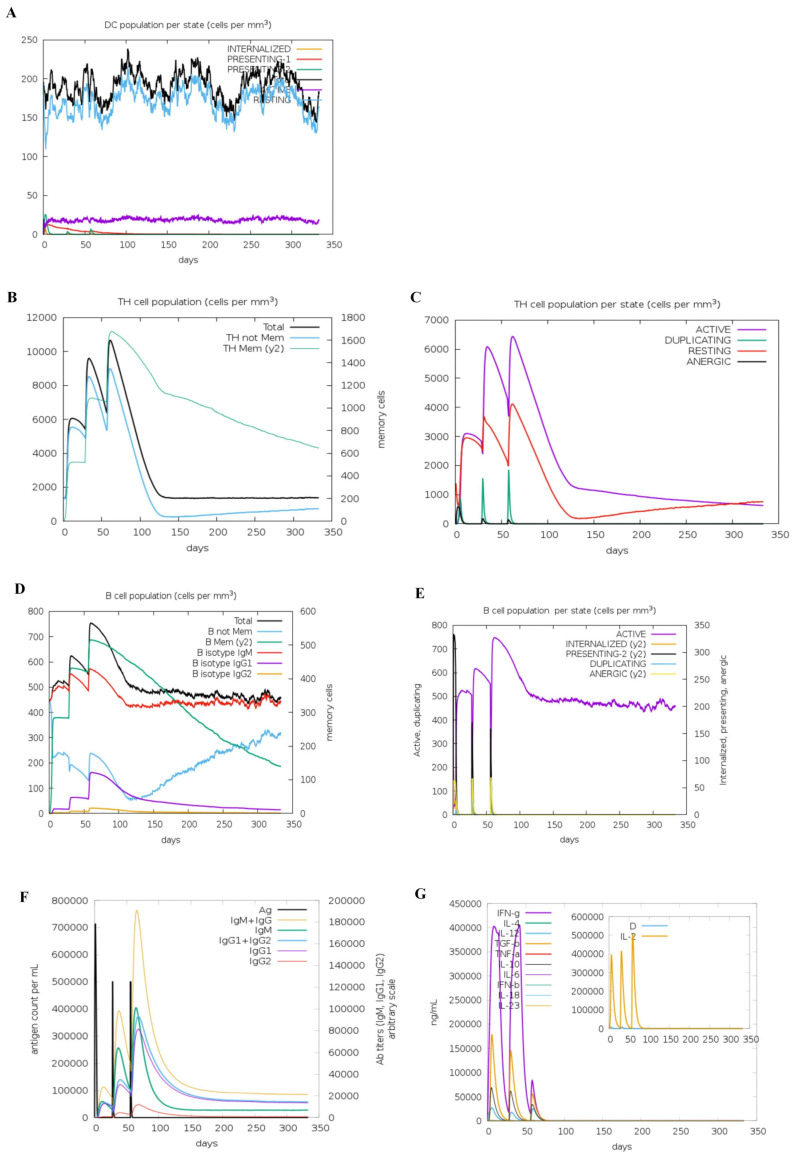
Immune simulation profiles of h-tFc-MEV vaccination. (**A**) DCs can present antigenic peptides on both MHC class-I and class-II molecules. The curves show the total number of bacteria broken down into active, resting, and internalized states and the corresponding. (**B**) The total number of TH cells. (**C**) TH cell population per state. (**D**) The total number of B cells. (**E**) B cell population per state. (**F**) Antibodies are subdivided according to isotype. (**G**) Concentrations of cytokines and interleukins. D in the inset plot is the danger signal.

**Figure 3 vaccines-13-00567-f003:**
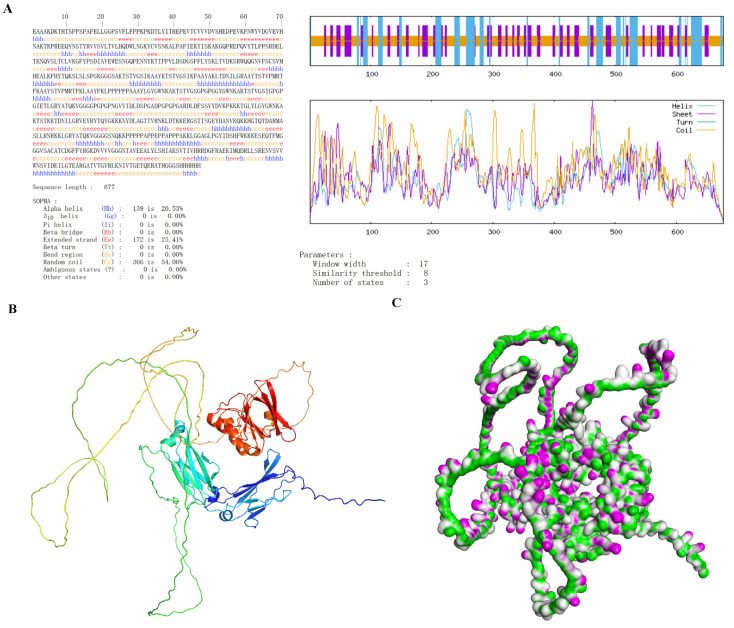
Structural characterization of h-tFc-MEV diagnostic antigen. (**A**) The secondary structure prediction, conducted using the SOPMA algorithm, indicates that α-helices comprise 20.53%, random coils account for 54.06%, and extended strands make up 25.41% of the structure. (**B**) The optimized tertiary structure, predicted by AF2, is visualized using Discover Studio software. (**C**) The hydrogen bond (H-bond) distribution profile is presented, with pink and green areas denoting donor and acceptor regions, respectively.

**Figure 4 vaccines-13-00567-f004:**
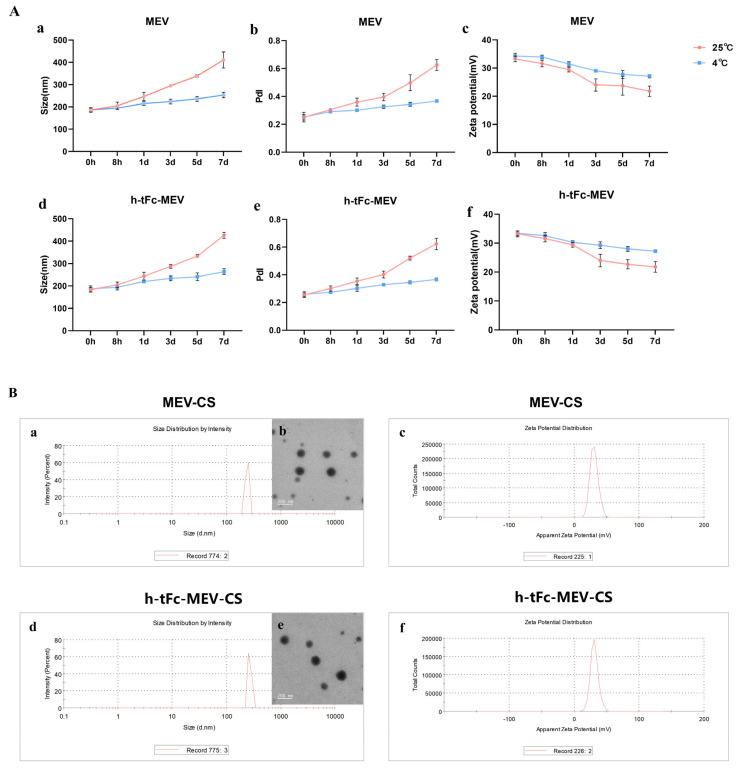
Structural and surface charge characterization of chitosan nanoparticle-encapsulated h-tFc-MEV and MEV. (**A**) Changes in particle size, PDI, and zeta potential of chitosan nanoparticles encapsulating proteins over 0–7 days (*n* = 3): (**a**) Particle size changes in MEV-loaded chitosan nanoparticles. (**b**) PDI changes in MEV-loaded chitosan nanoparticles. (**c**) Zeta potential of MEV-loaded chitosan nanoparticles. (**d**) Particle size changes in h-tFc-MEV-loaded chitosan nanoparticles. (**e**) PDI changes in h-tFc-MEV-loaded chitosan nanoparticles. (**f**) Zeta potential of h-tFc-MEV-loaded chitosan nanoparticles. (**B**) Characterization of chitosan nanoparticles: (**a**) Particle size distribution histogram of MEV-loaded chitosan nanoparticles. (**b**) TEM image of MEV-loaded chitosan nanoparticles. (**c**) Zeta potential diagram of MEV-loaded chitosan nanoparticles. (**d**) Particle size distribution histogram of h-tFc-MEV-loaded chitosan nanoparticles. (**e**) TEM image of h-tFc-MEV-loaded chitosan nanoparticles. (**f**) Zeta potential diagram of h-tFc-MEV-loaded chitosan nanoparticles.

**Figure 5 vaccines-13-00567-f005:**
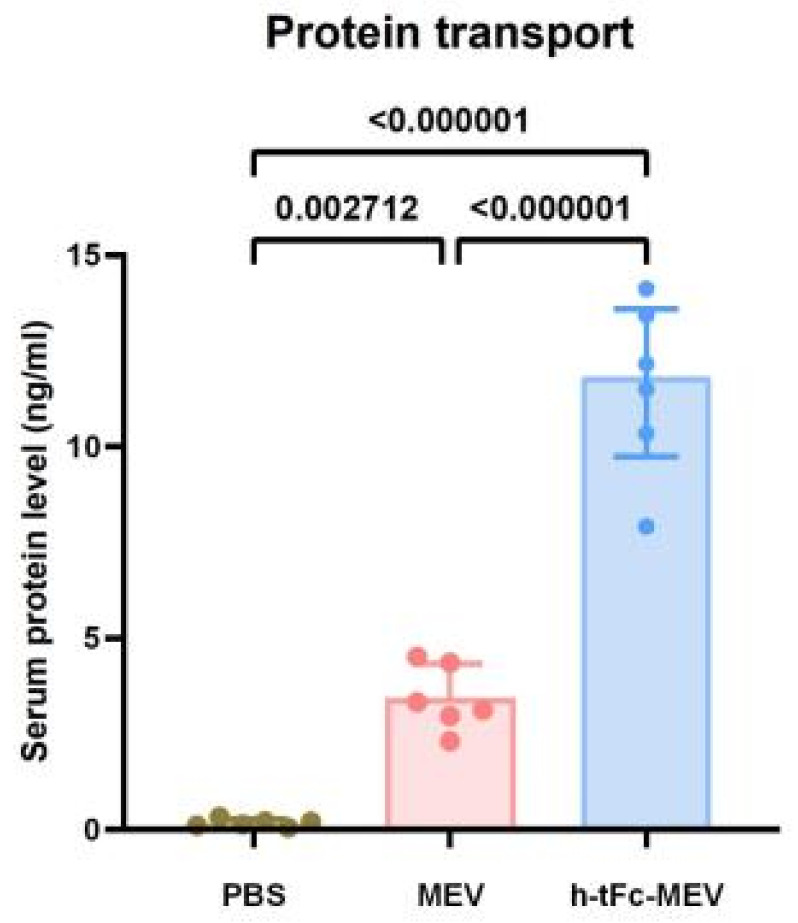
Quantitative analysis of serum antigen levels following oral administration of biotin-labeled h-tFc-MEV and MEV proteins. Balb/c mice (*n* = 6 per group) were administered 70 μg of either h-tFc-MEV or MEV protein via oral gavage, while the control group received an equivalent volume of PBS. Serum samples were collected 8 h post-administration, and target protein concentrations were determined using sandwich ELISA. The data are expressed as mean ± SEM (*n* = 6/group), with the heights of the columns representing group averages. Following the verification of data normality using the Shapiro–Wilk test, a univariate analysis of variance (ANOVA), along with the Tukey multiple comparison method, was employed to assess the differences among the groups. The statistical analysis was conducted using GraphPad Prism 9.0.0.

**Figure 6 vaccines-13-00567-f006:**
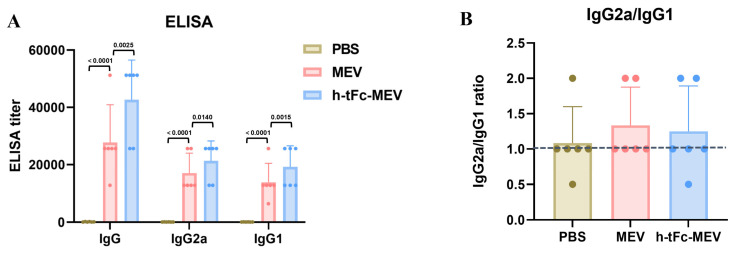
Antigen-specific humoral immunity and Th1/Th2 profiling in vaccinated mice. (**A**) Serum IgG, IgG2a, and IgG1 titers against MEV and h-tFc-MEV antigens were measured by ELISA at 14 days post-final immunization (*n* = 6/group). The data are presented as mean ± SEM (*n* = 6/group), with column heights representing group averages. (**B**) The specific IgG2a/IgG1 ratio is depicted, with the blue dashed line indicating the serum specific IgG2a/IgG1 ratio antibody titers in the PBS group. After verifying data normality via the Shapiro–Wilk test, the Kruskal–Wallis H test and Dunn post-hoc test (with Bonferroni correction) were employed. The statistical analysis was conducted using GraphPad Prism 9.0.0.

**Figure 7 vaccines-13-00567-f007:**
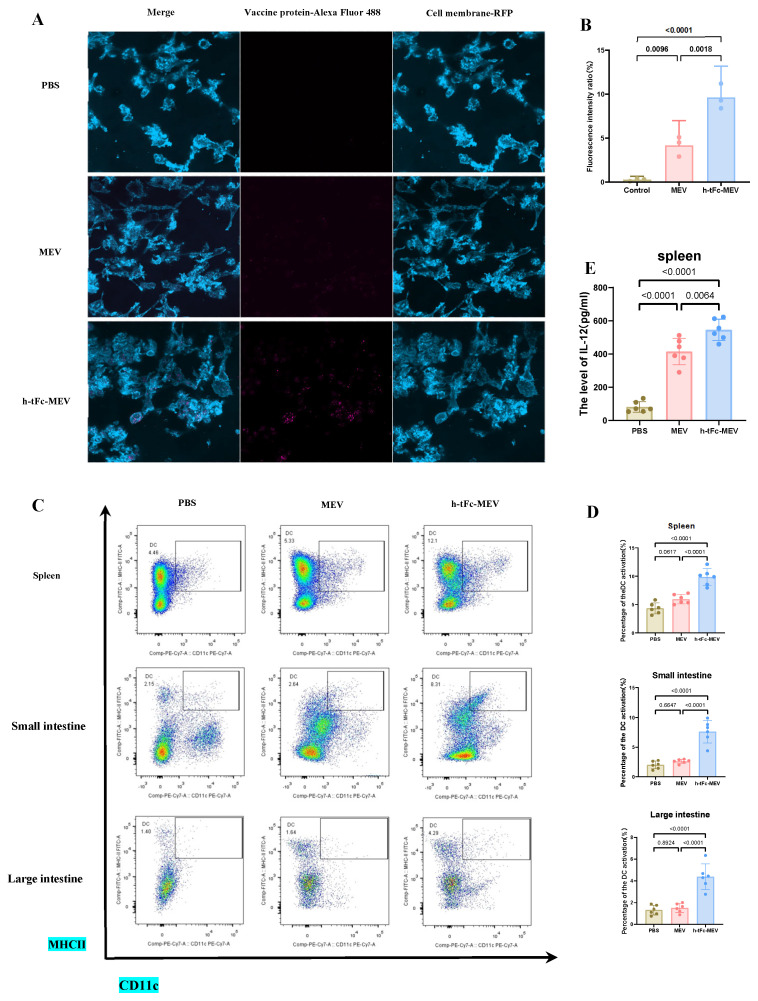
FcRn-targeted vaccine enhances antigen presentation and immune activation. (**A**) This panel displays immunofluorescence images of DCs treated with either h-tFc-MEV or MEV. (**B**) The semi-quantitative analysis of fluorescence intensity ratios (where h-tFc-MEV or MEV fluorescence intensity is normalized to total fluorescence) was conducted using ImageJ 1.54p (*n* = 3/group). (**C**) Representative flow cytometry plots depict mature DC populations (CD11c + MHC II+) in both the spleen and intestinal lamina propria, with the gating strategy detailed in Appendix A. (**D**) Quantification of the frequencies of mature DCs in the spleen and intestinal lamina propria is presented. (**E**) IL-12 levels in splenic homogenates were measured via ELISA. Data are presented as mean ± SEM (*n* = 6/group), with column heights representing group averages. Following the verification of data normality using the Shapiro–Wilk test, a univariate analysis of variance (ANOVA), along with the Tukey multiple comparison method, was employed to assess the differences among the groups. The statistical analysis was conducted using GraphPad Prism 9.0.0.

**Figure 8 vaccines-13-00567-f008:**
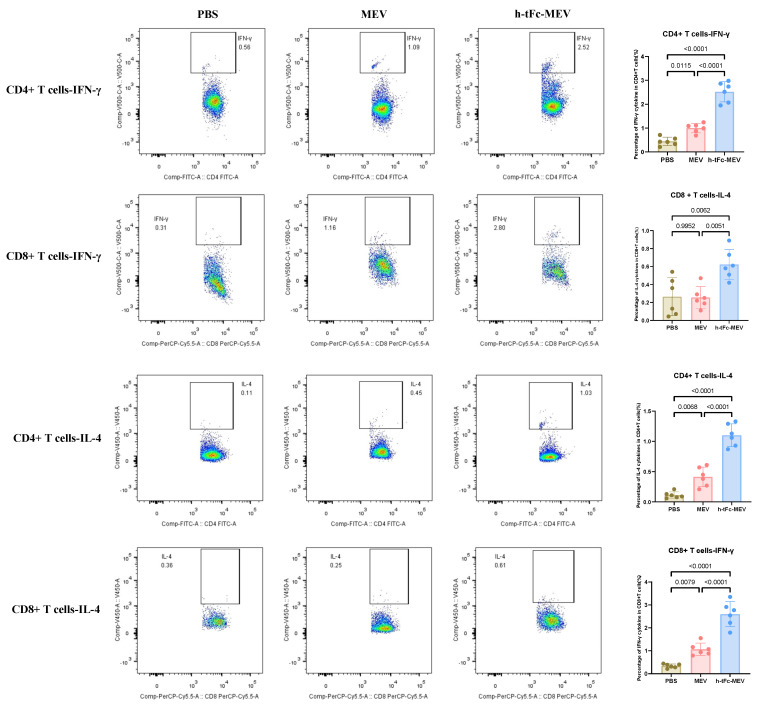
Flow cytometry analysis of changes in the ratio of IFN-γ + and IL-4 + CD4/CD8 + T cells in the spleen of mice 14 days after final immunization. The left panel presents a representative flow cytometry plot (refer to Appendix A for the gating strategy), while the right panel displays the statistical results of cell proportions in each group (mean ± SEM, *n* = 6). Following the verification of data normality using the Shapiro–Wilk test, a univariate analysis of variance (ANOVA), along with the Tukey multiple comparison method, was employed to assess the differences among the groups. The statistical analysis was conducted using GraphPad Prism 9.0.0.

**Figure 9 vaccines-13-00567-f009:**
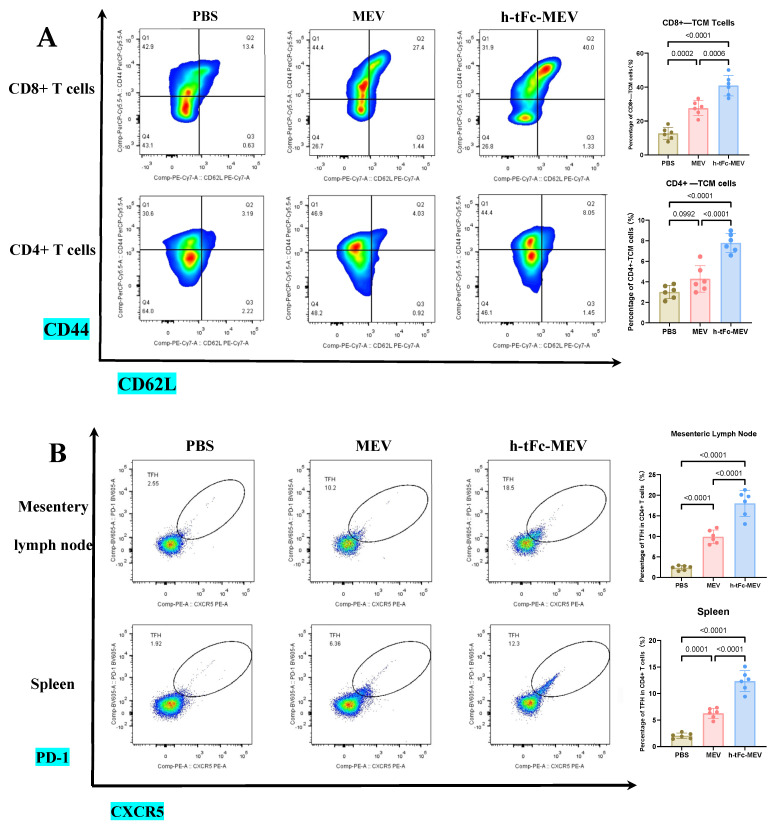
Immune profiling 6 months post−final immunization. (**A**) CD4^+^/CD8^+^ TCM cell proportions in spleen (flow cytometry; left: flow cytometry plot, right: group statistics). (**B**) TFH cell subsets in (**a**) mesenteric lymph nodes (MLN) and (**b**) spleen (flow cytometry; left: flow cytometry plot, right: group statistics). (**C**) GCB subpopulations in (**a**) MLN and (**b**) spleen (flow cytometry; left: flow cytometry plot, right: group statistics). (**D**) (**a**) Splenic IL-21 levels by ELISA and (**b**) serum antigen-specific IgG (IgG1/IgG2a) titers. Data expressed as mean ± SEM (*n* = 6/group; gating details in Appendix A). Following the verification of data normality using the Shapiro–Wilk test, a univariate analysis of variance (ANOVA), along with the Tukey multiple comparison method, was employed to assess the differences among the groups. The statistical analysis was conducted using GraphPad Prism 9.0.0.

**Figure 10 vaccines-13-00567-f010:**
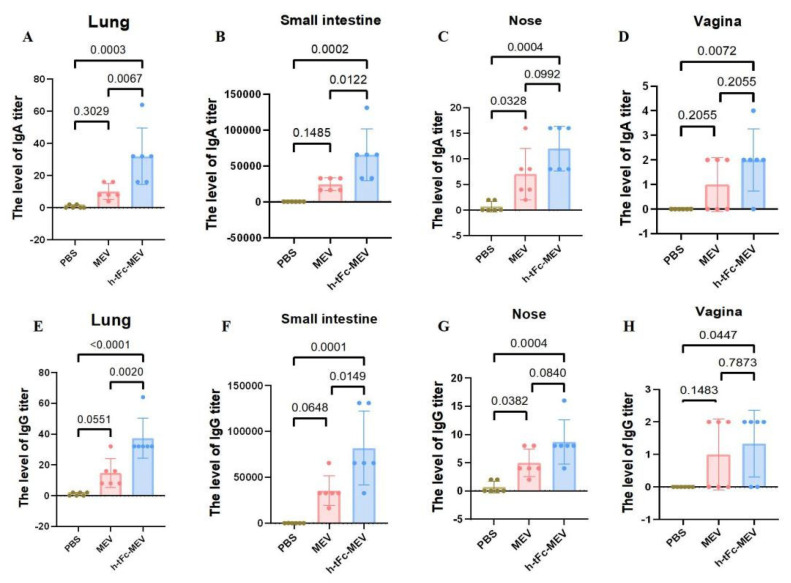
Mucosal antibody responses post h-tFc-MEV vaccination. (**A**) IgA levels in bronchoalveolar lavage fluid. (**B**) IgA levels in intestinal lumen lavage fluid. (**C**) IgA levels in nasal cavity lavage fluid. (**D**) IgA levels in vaginal tract lavage fluid. (**E**) IgG levels in bronchoalveolar lavage fluid. (**F**) IgG levels in intestinal lumen lavage fluid. (**G**) IgG levels in nasal cavity lavage fluid. (**H**) IgG levels in vaginal tract lavage fluid. The data are expressed as mean ± SEM (*n* = 6 per group). Following the verification of data normality using the Shapiro–Wilk test, a univariate analysis of variance (ANOVA), along with the Tukey multiple comparison method, was employed to assess the differences among the groups. The statistical analysis was conducted using GraphPad Prism 9.0.0.

**Figure 11 vaccines-13-00567-f011:**
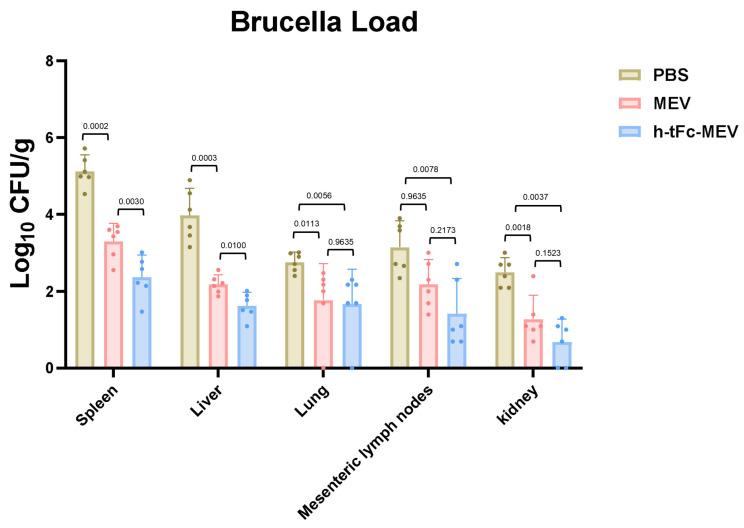
Bacterial colonization profiles in reticuloendothelial organs post-challenge. Three experimental groups (PBS control, MEV vaccine, and h-tFc-MEV vaccine) underwent triple oral immunization, followed by a *Brucella melitensis* challenge (10 log10 CFU) at two weeks post-immunization. Tissue bacterial loads in the spleen, liver, lungs, mesenteric lymph nodes (MLN), and kidneys were quantified via plate culture at 14 days post-challenge. The data are expressed as mean ± SEM (*n* = 6/group). Following the verification of data normality using the Shapiro–Wilk test, a univariate analysis of variance (ANOVA), along with the Tukey multiple comparison method, was employed to assess the differences among the groups. The statistical analysis was conducted using GraphPad Prism 9.0.0.

**Table 1 vaccines-13-00567-t001:** The physical and chemical properties of protein vaccine candidates.

Protein	Omp25	VirB10
Accession number	Q45321	Q8YDZ0
Amino acid number	213aa	380aa
Subcellular localization	Cell outer membrane	Cell outer membrane
Antigenicity	0.8164	0.6906
Allergenicity	non-allergen	non-allergen
Stability	23.00	36.52
Hydrophilicity	−0.317	−0.155

Note: Stability < 40.00 is considered stable; if the hydrophilicity is negative, it means hydrophilic, and the smaller the negative value is, the more hydrophilic it is. Antigenicity > 0.4 is considered a protective antigen.

**Table 2 vaccines-13-00567-t002:** The physical and chemical properties of h-tFc-MEV vaccine.

Protein	h-tFc-MEV
Molecular weight	71.54 kDa
Formula	C3198H5027N883O957S12
Theoretical pI	9.59
Antigenicity	0.6878
Allergenicity	non-allergen
Stability	stable
Hydrophilicity	−0.577
Solubility	Good (Appendix A)

## Data Availability

Data sharing is not applicable to this article. Requests to access datasets should be directed to the corresponding author.

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
