# Peer review of "FcRn-Driven Nanoengineered Mucosal Vaccine with Multi-Epitope Fusion Induces Robust Dual Immunity and Long-Term Protection Against Brucella"

_vaccines, 2025, doi:10.3390/vaccines13060567_

Round 1
Reviewer 1 Report
Comments and Suggestions for Authors
The authors have attempted a good approach of inducing mucosal immunity towards Brucella antigens through the use of a chitosan delivery vehicle. it is interesting to see that this approach induces a mucosal response.
Minor comments
Fig4a: With the Electron micrographs it is clear that the nanoparticles are lower than 200nm, but the authors should show DLS histograms to confirm the size distribution of the nanoparticles.
Fig 6b should show error bars
Supplemental
In the graphical representation , the disruption in the C1q ablation motif(Red) is not clear whether there is mutagenesis or deletion of the particular amino acids KEYKCK-KAYACA, reprentation shows KYCV?
In the western blot for the HIS tag in suppl Fig 5, could the authors shed light on why there are non specific bands in the same lane as the anti-his tag antibody, which ideally should be specific and should result in just one single band.
Reviewer 2 Report
Comments and Suggestions for Authors
These are my comments, a few major and most minor.
Line 46-47, "evolutionary pressure that favors the development of antimicrobial resistance in Brucella populations due to prolonged antibiotic use in livestock", this has nothing to do with vaccines.
Line 93-94, "To overcome the limited efficacy of conventional vaccines against Brucella’s intracellular persistence", add references to support this statement.
Line 311, Protocol?
Line 370, 1×10^10?
Table 2, use 71.54 kDa, not 71.54kda
Figure 2, increase size of all graphs
Line 567-568, "proteins were purified under native conditions via Ni-NTA affinity chromatography with a linear imidazole gradient (elution at 150 mM)", the methodology states no gradient with elution at 300 mM.
Line 581, "with average particle diameters below 200 nm", what was the polidispersity index (PdI)?
Line 581-583, "which satisfies the essential size criterion (<500 nm) for effective penetration through intestinal mucus", add references to support this statement.
Line 583-584, "zeta potentials of 583 +7.27 ± 5.54 mV and +4.44 ± 4.49 mV respectively", the methodology does not establish under what conditions were these potentials measured. Were the particles dispersed in 2 mM NaCl? If the measured potentials are true and the particles are stabilized electrostatically, they are unstable. Determine stability (measure size, PdI, and zeta potential over time) over a period of several days at 4 C and room temperature. For this, you must disperse the particles in your vehicle (PBS), and then dilute the sample for the zeta potential measurement.
Line 586-587, "Notably high protein loading efficiencies of 65.12% (h-tFc-MEV) and 61.42% (MEV) were achieved", the methodology section is not clear with respect to how these efficiencies were determined. A mass balance was applied to determine an equilibrium concentration? Or the particles were washed several times, measuring protein concentration in the washing solution, and a mass balance was applied to determine strongly adsorbed or entrapped protein?
Line 604, PBS proteins?
Section 3.11, a word regarding the fact that no significant differences are observed when comparing MEV and h-tFc-MEV (Figure 6A).
Figure 10, a word again regarding the comparison of MEV and h-tFc-MEV, no significant differences are observed.
Figure 11, a proper statistical analysis must be performed
Reviewer 3 Report
Comments and Suggestions for Authors
The idea of the article is interesting, providing a novel idea for a vaccine. In general, it is well written, but many things should be amended before it is suitable for publication.
First, the article has no Ethical statement or welfare committee-approved number or reference, which is unacceptable for a scientific article involving animals.
The abstract lacks a clear explanation, even if the vaccine is designed for human or animal species. The focus or the target Brucella species is unclear until the discussion ends. The abstract should provide a clear summary of the article, enough to give an idea about the article's information.
All websites and apps used should be identified with the time when they were accessed (web), version, etc. It is not homogeneously done throughout the text.
Line 232-233. Even when a third party does the manufacturing, it is recommended to add a brief explanation.
Commercial name, country and city should be added to the products used. It is not coherent through the text, being versions such as "Sigma-Aldrich, I6758", "Malvern Instruments, Worcestershire, UK", or "Sigma-Aldrich, Germany".
Terms in Latin, such as "in vivo", should be in italics.
Once a term is explained and an abbreviation provided, it should not be repeated, such as in "phosphate-buffered saline. "
References should be checked; for example, Reference 42 does not refer to what was expressed in the text.
species can have both smooth and rough strains
Figure 1 may be in the introduction, not the Materials and Methods.
What do the authors refer to: antigens 39,40 (line 498), neutralisation 41 (Line 499), macrophages 42 (Line 503) or host defence 43 (line 505)? It looks like they try to add references, but it is unclear. If this were the case, they should seriously check the references.
Structure: Only the results section should be presented. Any further comments or explanations should be in the discussion. E.g. Section 3.14
The figure reference should be minimal to explain the text. Some figures have a one-page explanation, which is too long for a scientific paper, e.g. figure 9.
The sentence in lines 767-768 should be the last sentence of the introduction instead of being here.
Please check the writing, e.g. Line 793 "2x1010 or 2e10" instead of "2×1010"
Use a homogeneous nomenclature, not 1.74-log10 CFU/g or 2×1010 CFU or 1×10^10 CFU.
The paragraph is controversial, lines 853-856. "Needle-free mucosal vaccination demonstrates significant advantages over conventional parenteral administration, particularly through its ability to eliminate requirements for specialized medical personnel and mitigate needle-stick contamination risks, thereby substantially improving vaccination compliance". Even when needles were not involved, specialised personnel should administer all vaccines.
Round 2
Reviewer 2 Report
Comments and Suggestions for Authors
The authors have addressed all my concerns. Dimensions and units must be separated, the authors have a mess in this regard, they use 0h, 0 h, 10kDa, 10 kDa. Use only 0 h and 10 kDa.
Author Response
The authors have addressed all my concerns. Dimensions and units must be separated, the authors have a mess in this regard, they use 0h, 0 h, 10kDa, 10 kDa. Use only 0 h and 10 kDa.
Reply:
Dear Reviewer,
Thank you for bringing this formatting inconsistency to our attention. We have carefully reviewed the manuscript and corrected all instances of dimensions and units. Now, throughout the text, we consistently use "0 h" (with a space between the value and the unit) and "10 kDa" (with a space between the numerical value and the unit abbreviation).
We appreciate your vigilance in helping us maintain clarity and consistency in our scientific communication.
Modify the position: Line 287; 300; 310; 362; 364; 353; 633; 640; 649; 727
Thank you for your guidance in improving the clarity and quality of our work. Have a great day!
Sincerely,
The Authors